# A feedback mechanism converts individual cell features into a supracellular ECM structure in *Drosophila* trachea

**Arzu Öztürk-Çolak[1,2], Bernard Moussian[3,4], Sofia J Araújo[1,2]\*, Jordi Casanova[1,2]\***

[1]Institut de Biologia Molecular de Barcelona, Parc Cientific de Barcelona, Barcelona, Spain; [2]Institut de Recerca Biomedica de Barcelona, Parc Cientific de Barcelona, Barcelona, Spain; [3]Animal Genetics, Interfaculty Institute for Cell Biology, University of Tuebingen, Tuebingen, Germany; [4]Institute of Biology Valrose, Faculté des Sciences, University of Nice Sophia Antipolis, Nice, France

**Abstract** The extracellular matrix (ECM), a structure contributed to and commonly shared by many cells in an organism, plays an active role during morphogenesis. Here, we used the *Drosophila* tracheal system to study the complex relationship between the ECM and epithelial cells during development. We show that there is an active feedback mechanism between the apical ECM (aECM) and the apical F-actin in tracheal cells. Furthermore, we reveal that cell-cell junctions are key players in this aECM patterning and organisation and that individual cells contribute autonomously to their aECM. Strikingly, changes in the aECM influence the levels of phosphorylated Src42A (pSrc) at cell junctions. Therefore, we propose that Src42A phosphorylation levels provide a link for the ECM environment to ensure proper cytoskeletal organisation.

**\*For correspondence:** sarbmc@ibmb.csic.es (SJA); jcrbmc@ibmb.csic.es (JC)

**Competing interests:** The authors declare that no competing interests exist.

## Introduction

*"The anatomically internal lining of the trachea consists of a chitinous layer which presents what is usually termed a 'spiral thickening', but whose form is really that of a helicoid. (...) then we have to explain just how it is that the cells of the tracheal epithelium can cooperate unconsciously so as to form a helicoid thickening continuous from one end of the trachea to another, especially since each cell produces not merely one section of the continuous filament, but several parallel sections of unequal length."* (**Thompson, 1929**)

The morphology of organs, and hence their proper physiology, relies to a considerable extent on the extracellular matrix (ECM) secreted by their cells. On the one hand, the ECM has a key role in cell signalling events, either acting as a storage compartment for growth factor molecules or by modifying and/or stabilising such growth factors. On the other hand, the ECM provides an environment with a variable degree of stiffness that, until recently, has been considered a physical support to ensure the arrangement of softer parts of body organs. However, increasing evidence indicates that the ECM not only provides a passive contribution to organ shape but also impinges on the cell behaviour and genetic programmes of the organ (**Daley and Yamada, 2013**). The ECM is emerging as a direct modulator of many aspects of cell biology, rather than as a mere physical network that supports cells (**Hynes, 2014**). But how does the ECM influence cell biology and which molecules are involved?

There are many types of ECMs, ranging from cell walls and basement membranes to highly specialised structures such as tendons and shells. The assembly of these structures requires a complex set of diverse events involving protein secretion, as well as the modification of these proteins. In spite of

**eLife digest** Animal cells can secrete proteins and molecules into the space around them to create a support they can attach to. This structure – known as the extracellular matrix – comes in various forms and can help to shape tissues or influence the way in which cells behave. Inside cells, filaments made of a protein called actin also provide structural support.

In fruit fly larvae, "tracheal" cells create a network of tubes that will form the airways of the adult fly. Once this network is complete, these cells secrete the materials to make an extracellular matrix in the internal (apical) surface of the tubes. This matrix has a series of spiralling ridges made from a molecule called chitin. These ridges run along the tubes, spanning several cells and providing the mechanical strength needed to keep the airways open.

The ridges appear to form through a co-ordinated effort between the cells, and recent studies suggest that actin filaments may be involved in this process. Here, Öztürk-Çolak et al. investigate this idea further by carrying out a detailed analysis of the relationship between the extracellular matrix and the tracheal cells as the airways develop. The experiments reveal that rings of actin filaments form on the apical side of tracheal cells before the ridges appear. These rings generate regular folds in the membrane that surrounds each tracheal cell and are required for an enzyme to accumulate in the cells. This enzyme produces chitin, leading to its deposition in stripes above the actin rings.

Further experiments show that the junctions between cells play an important role in organising the pattern of the extracellular matrix. The active form of a protein called Src42A – which is known to regulate the way actin filaments are organized inside cells – accumulates at these junctions. Excessive Src42A activity in tracheal cells alters the networks of actin filaments and disrupts the formation of the matrix. Öztürk-Çolak et al. also find evidence of a "feedback" mechanism, in which the presence of chitin reduces the activity of Src42A to maintain the correct patterning of actin.

These findings reveal that actin and junctions between cells play a central role in co-ordinating the formation of the extracellular matrix in fruit fly airways. The next challenge will be to understand which proteins and other molecules are involved in the process that allows the extracellular matrix to communicate with the cells.

the relevance of ECMs, it is not yet well understood how they are generated. In particular, a fascinating aspect of ECMs is the fact that they usually constitute a structure contributed to and shared by many cells. But how do individual cells participate in the generation of a supracellular ECM with an overall common organisation that overruns cellular borders?

We have addressed these issues by studying the apical ECM (aECM) of *Drosophila melanogaster* trachea, the insect respiratory system. Once the different branches of the tracheal system have been established to cover the overall embryonic body, tracheal cells begin to secrete the components of a chitin-rich aECM that lines up the lumen of the tracheal tubes and can be visualised by the incorporation of chitin-binding probes (*Moussian et al., 2005*). A distinctive feature of this aECM are taenidial folds, a series of cuticle ridges that compose a helical structure running perpendicular to the tube length along the entire lumen (*Wigglesworth, 1990*). Taenidia are believed to confer mechanical strength to the tubes and have been compared to a coiled spring within a rubber tube (*Thompson, 1929*) or to the corrugated hose of a vacuum cleaner (*Manning and Krasnow, 1993*). From the very first descriptions, it was noticed that taenidia are unaffected by the presence of cell boundaries (*Thompson, 1929*), thereby indicating that they are a supracellular structure and suggesting a substantial degree of intercellular coordination. More recently, it has been reported that taenidial organisation correlates with that of the apical F-actin bundles in underlying cells—the formation of these bundles preceding the appearance of taenidia (*Matusek et al., 2006*; *Kondo et al., 2007*). However, the relationship between these bundles and taenidia is still poorly understood. In addition, physical modelling has recently revealed that the interaction of the apical cellular membrane and the aECM determines the stability of biological tubes (*Dong et al., 2014*), thus generating more questions about how this interaction occurs.

Here, we report that there is a dynamic relationship between sub-apical F-actin and taenidial folds during tracheal lumen formation. We show that cell-cell junctions participate in organising F-actin bundles and the taenidial fold supracellular aECM and that this chitinous aECM contributes to regulating F-actin organisation in a two-way regulatory mechanism.

## Results and discussion

### Time course of actin ring and taenidial fold formation

In order to obtain a detailed framework of taenidial fold formation during embryonic development, we began by performing a detailed analysis of the timing of taenidial formation. We focused on the main branch of the trachea, the dorsal trunk (DT), where taenidia are more conspicuous. It is worth mentioning that, prior to taenidial fold formation, a transient chitin filament is formed inside the tracheal lumen. This filament has been postulated to regulate tube length and diameter expansion (*Tonning et al., 2005*; *Moussian et al., 2006a*; *Luschnig and Uv, 2014*). As this filament is a transient structure, its appearance in and disappearance from the lumen of the DT is a useful landmark to precisely stage embryos. Taenidia began to be detectable by late stage 16 when the chitin filament was still present in the tracheal lumen (*Figure 1A*). Optical section analysis showed that taenidia develop at the more external luminal sections, while the chitin filament lies in a central position inside the lumen (*Figure 1A*). From early stage 17, a stage when the luminal chitin fibre is already absent (*Moussian et al., 2006b*), taenidia became increasingly more prominent (*Figure 1E*). As mentioned above, taenidial folds were organised as spiral rings that span many distinct cells (*Figure 1L*).

Given the close correlation between taenidia and the rings of actin bundles (*Matusek et al., 2006*), we next analysed the developmental time course of these two structures in the same embryo. For this purpose, we used fluostain and phalloidin to visualise chitin and F-actin, respectively (*Moussian et al., 2005*; *Araújo et al., 2005*). At early stage 16, when taenidia were not yet detectable, we distinguished some actin rings in the cells of the DT (*Figure 1G*). It is noteworthy that the first actin rings to appear in the trachea were those corresponding to the fusion cells, which are not related to taenidia but instead to the fusion between the lumen of adjacent segments of the DT (*Lee and Kolodziej, 2002*). Shortly after, actin rings in other cells were detected throughout the length of the DT, but these were much weaker than those present in the fusion cells (*Figure 1G''*). At this stage, remnants of the chitin filament were still detectable but taenidial folds were not (*Figure 1G'*). As the chitin filament faded away and taenidia became more visible, the actin rings became more defined and prominent (*Figure 1H*; for a more detailed time course of actin ring and taenidial fold formation see *Figure 1—figure supplement 1*).

To further study the functional link between actin rings and taenidial folds, we addressed whether these two structures are generated in frame. Maximum projection from confocal planes obtained from embryos stained with phalloidin and fluostain supported this notion (*Figure 1H*), similarly to what has been shown upon F-actin labelling in fluorescent and DIC images at larval stages (*Figure 1I, J* and [*Matusek et al., 2006*]). In embryos, it is more difficult to reach a clear conclusion from confocal projection images, as the two structures are in different planes. However, transmission electron microscopy (TEM) images of the embryonic tracheal DT show the presence of electron dense structures in the cytoplasm underneath the taenidia (*Figure 1K*, open arrow). These densities are likely to be cross-sections of actin filaments and not vesicles or microtubules because of their size. Whereas microtubules appear as circular structures of about 25 nm in diameter and vesicles are generally 5–20 times larger, actin filaments are only about 7 nm in diameter (*Grazi, 1997*). Taken together these observations, lead us to conclude that actin filaments and taenidia are 'in-frame'.

### Impairment of apical actin rings similarly reshapes taenidia

Alteration of the tracheal apical F-bundles by mutants in genes encoding actin polymerisation proteins also cause defects in taenidial arrangement (*Matusek et al., 2006*). Such is the case for mutants for *tarsal-less* (*tal*), also known as *polished rice* (*pri*), a gene transcribed into a polycistronic mRNA that contains short ORFs encoding 11 or 32 amino acid-long peptides (*Kondo et al., 2007*; *Galindo et al., 2007*). Interestingly, *tal/pri* is essential for the formation of actin bundles that prefigure two chitin structures, namely denticles in the embryonic cuticle and taenidia in the trachea (*Kondo et al., 2007*). To further address the contribution of tracheal actin bundles to the

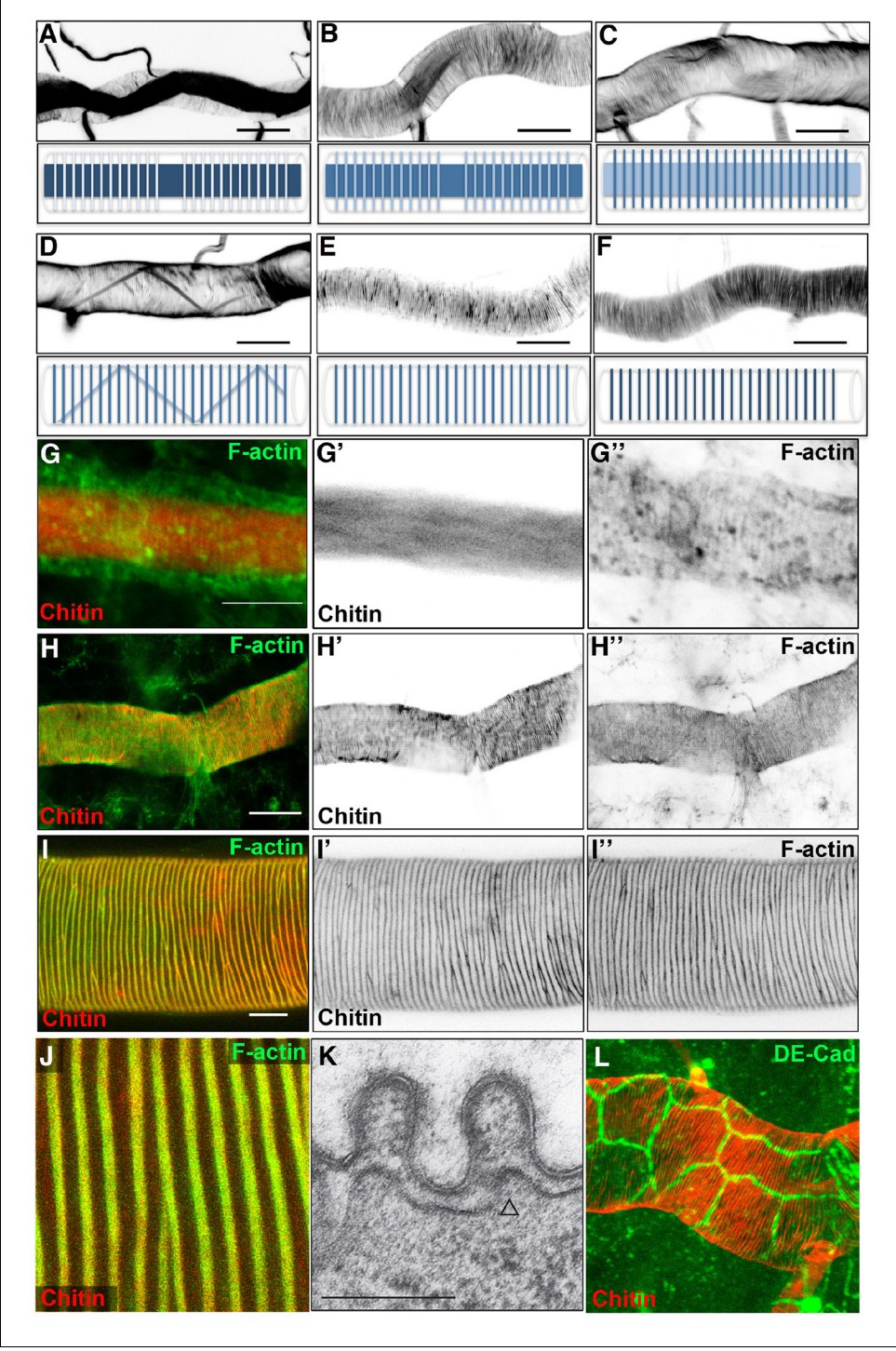

**Figure 1.** Dynamics of taenidial fold and actin ring formation. (A–F) Dorsal Trunk detail of wild-type embryos stained with fluostain to label the chitin structures. Maximum projections of confocal Z sections showing the dynamics of intraluminal chitin filament and taenidial folds during late stages of embryonic development. Chitin structures are schematically represented under each image. <u>Chitin filament</u>: at late stage 16, intraluminal chitin filament is thick and dense (A); as the embryo develops, it becomes less and less dense (B, C) until it turns into a thin chitin fibre that runs in zigzags along the tube diameter (D); in the last steps of embryogenesis, the intraluminal chitin filament is completely cleared from the lumen (E, F). <u>Taenidial folds</u>: at late stage 16, taenidial folds are newly formed and thin (A); as the taenidial folds become thicker, it is apparent that the taenidial folds at fusion points are not formed yet (B); later, the taenidial folds at fusion points are also formed which generates a

*Figure 1 continued on next page*

*Figure 1 continued*

continuous taenidial structure along the tube (**C**); in the final steps, as the intraluminal chitin filament is cleared from the lumen, the taenidial folds reach their their most mature form (**D–F**). Scale bars = 10 μm. (**G–J**) wild-type embryonic (**G–H**) and 3rd instar larval (**I–J**) trachea stained with fluostain (red) and phalloidin (green) showing taenidial folds and F-actin bundles together (**G, H, I, J**) or separately (**G', G'', H', H''** and **I', I''**). F-actin organisation in structures perpendicular to the main tube axis occurs at stage 16 prior to taenidial fold appearance (**G**). When taenidia become apparent (**H**), they are positioned over the actin bundles and this co-localisation continues throughout larval stages (**I, J**). Scale bars = 10 μm. (**K**) TEM detail of wild-type DT taenidia, the open triangle points to actin filament cross-sections (diameter around 7 nm). Scale bar 250 nm. (**L**) Detail of DT showing apical cell borders (labelled by anti-DE-Cad in green) and how taenidia (labelled by fluostain in red) span continuously beyond cell-cell borders from one cell to the other.

The following figure supplement is available for figure 1:

**Figure supplement 1.** Time course of actin ring and taenidial fold formation.

arrangement of taenidia, we examined these two structures in the same mutant trachea to assess whether they are strictly correlated. In most *tal/pri* mutant embryos, the tracheal F-actin bundles formed but they were misoriented and did not follow the ring distribution found in the wild-type (*Figure 2B,D*); in some extreme cases, the bundles were completely disorganised. In all the cases studied, we found taenidia to be organised along the same pattern as the F-actin bundles, either running parallel to the tube axis when actin bundles were oriented in this way (*Figure 2C*), completely disorganised, or completely misshapen in twisted tracheal tubes when F-actin fibres were aligned in a twisted manner.

Similarly to *tal/pri*, mutants in the gene coding for the DAAM formin also disrupt tracheal actin rings and taenidia; this is also the case for the mutants in the gene coding for the Btk29A non-receptor tyrosine kinase, which interacts genetically with DAMM (*Matusek et al., 2006*). In both mutants, we found heterogeneous patterns of F-actin bundling in the same trachea, with stretches of perpendicular bundles followed by stretches of parallel bundles (*Figure 2F, H*). In support of the close relationship between F-actin bundles and taenidia, we found the latter to reproduce the stretches of parallel and perpendicular orientation of the former (*Figure 2E,G*). Of note, actin bundles were harder to observe in mutants for DAAM as they were much thinner than in the wild-type (*Figure 2F*).

We extended our analysis to other genes such as *singed* (*sn*) and *forked* (*f*), which code for a fascin and an actin-binding protein, respectively, both expressed in tracheal cells (*Okenve-Ramos and Llimargas, 2014*) and also required for other cuticle structures such as adult bristles (*Overton, 1967*). However, neither appear to be involved in the tracheal actin rings as these structures formed normally in *sn* and *f* mutants and also in *sn;f* double mutants (data not shown). All together, these results indicate that tracheal actin rings are not only required for proper organisation of taenidia in larvae (*Matusek et al., 2006*) but also appear to play an instructive role in determining their initial formation during embryonic stages.

## Tracheal actin rings relate to the spatial distribution of the chitin synthase

How could tracheal actin rings instruct the positioning of taenidia? EM images have identified the accumulation of electron-dense material at the base of the taenidia that is thought to correspond to the chitin synthesis complex (*Uv and Moussian, 2010*) and that appears to be located at the top of actin fibres (*Moussian et al., 2006b*). Thus, the tracheal actin rings may instruct taenidial fold shaping by dictating the sites of chitin production. In the *Drosophila* trachea, this production is carried out by the membrane-inserted chitin synthase encoded by the *krotzkopf verkehrt* (*kkv*) gene (*Ostrowski et al., 2002*). The Kkv chitin synthase enzyme catalyses the linkage between N-acetyl-glucosamines supplied in the cytoplasm and extrudes the polymers across the plasma membrane (for a review see [*Moussian et al., 2006b*]). Therefore, we analysed Kkv distribution in tracheal cells. For this purpose, we made use of a recently generated transgenic strain carrying a *kkv-GFP*-tagged construct under the control of the UAS promoter sequences (*Moussian et al., 2015*). When driven by the trachea-specific *btlGal4* driver, it rescued the tracheal phenotype of the *kkv* mutants (our own

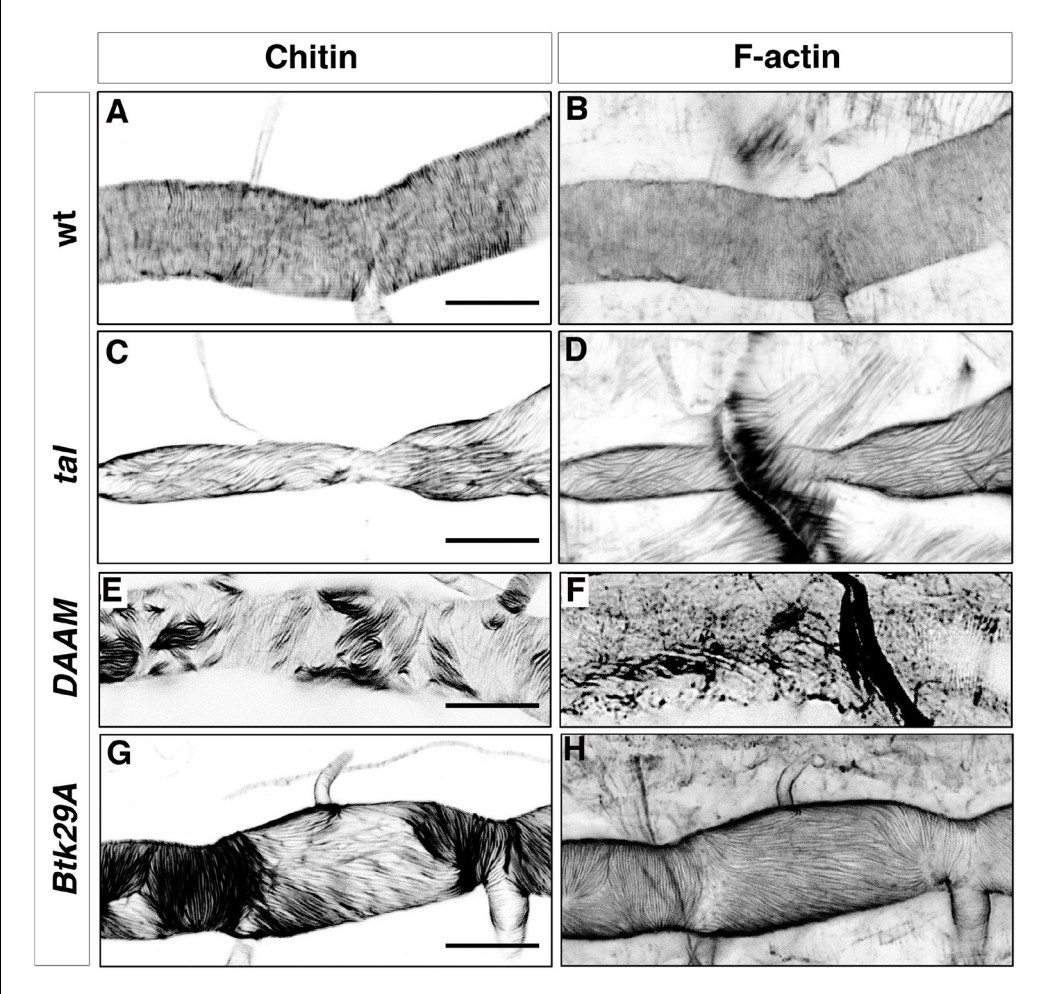

**Figure 2.** Taenidial folds and F-actin rings in *tal/pri*, *DAAM,* and *Btk29A* mutants. Wild-type (**A, B**), tal/*pri* (**C-D**), DAAM (**E-F**), and Btk29A (**G-H**) mutant embryos stained with fluostain (**A, C, E, G**) to label taenidial folds and phalloidin (**B, D, F, H**) to label F-actin bundles. Both taenidial folds and F-actin bundles run perpendicular to the tube axis in wild-type embryos (**A, B**) while in most of the *pri* mutants they are parallel to the tube axis (**C, D**). In *DAAM* (**E-F**) and *Btk29A* (**G-H**) mutant embryos, taenidial folds and actin bundles are hardly detected and when so they appear mis-oriented, running both parallel and perpendicular to the tube axis. In all panels anterior is to the left and scale bars =10 μm.

results and [*Moussian et al., 2015*]), thereby indicating that the Kkv-GFP-tagged protein is fully functional and localises properly.

KkvGFP showed a highly punctated accumulation in tracheal cells, which makes it difficult to distinguish clear spatial patterns. However, we did detect linear arrangements perpendicular to the tube length that resembled the actin rings detected by phalloidin and the chitin rings detected by fluostain (*Figure 3B and C* and *Figure 3—figure supplement 1*). We have quantified the number of times these KkvGFP dots overlap with the fluostain rings and observed that they are indeed more frequent within each ring (on average, 25 dots per taenidium in contrast to 15 dots outside, n = 24, *Figure 3—figure supplement 1E*). Interestingly, we also found that these KkvGFP dots are larger when they overlap with the fluostain rings than when they are present outside (*Figure 3—figure supplement 1D*). These findings support the notion that actin organisation, which dictates taenidial fold organization, participates in the distribution of the Kkv chitin synthase. Interestingly, we did not observe the same arrangements upon driving *kkvGFP* expression in a *tal/pri* mutant background (100%, n = 30, *Figure 3D*). Instead, we found that Kkv dots are more aligned with the disrupted

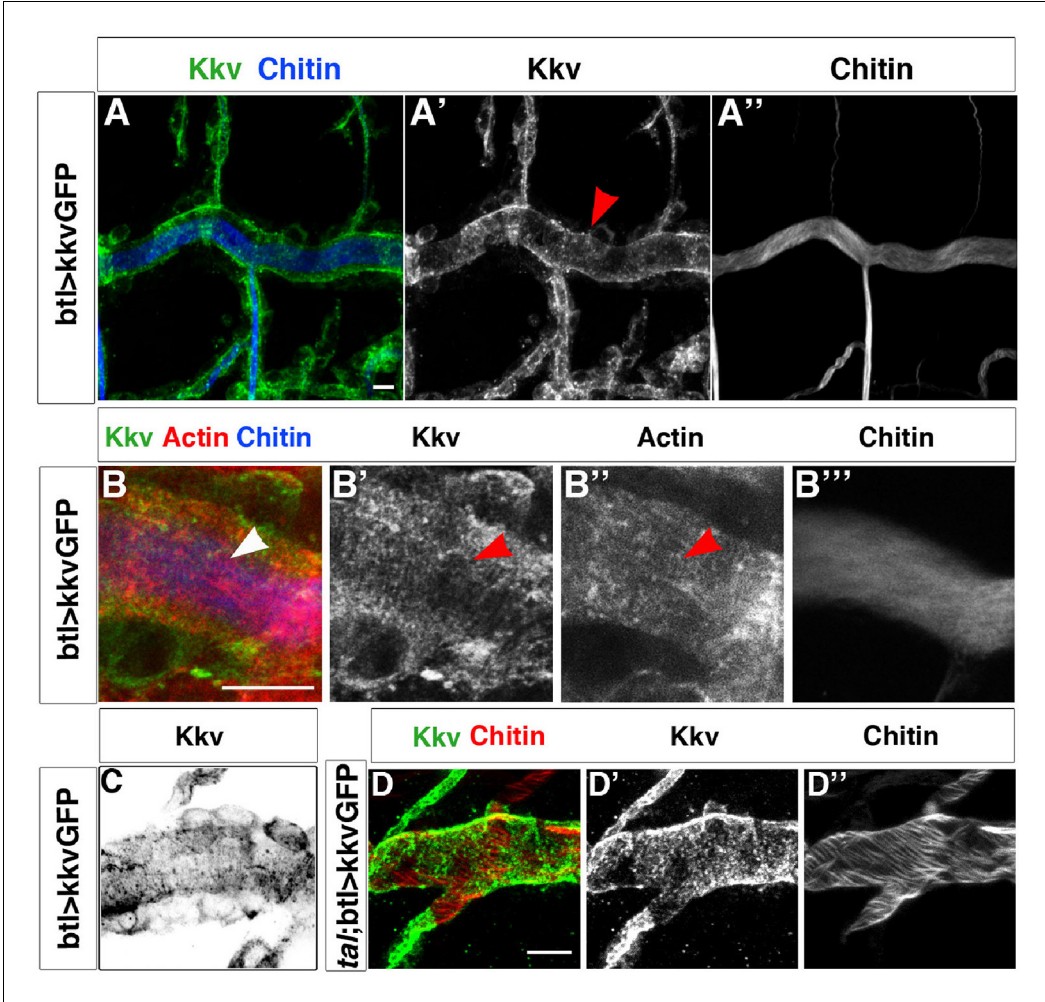

**Figure 3.** Kkv co-localises with F-actin rings during tracheal maturation. (**A**) Detail of stage 16 wild-type embryos expressing a *kkvGFP* transgene in tracheal cells and stained with fluostain to reveal the chitin filament; Kkv is detected in rings that resemble the F-actin rings (arrowhead in **A'**). (**B**) DT zoomed detail of stage 16 wild-type embryos expressing a *kkvGFP* transgene in tracheal cells and stained with fluostain and phalloidin to reveal the chitin filament and the F-actin rings (arrowheads). (**C**) Detail of a stage 17 embryo expressing *kkvGFP* in tracheal cells, showing the localisation of Kkv in rings throughout the length of the tube. (**D**) Detail of stage 17 *tal/pri* embryos expressing a *kkvGFP* transgene and stained with fluostain to reveal the chitin aECM; in *tal/pri* mutants, Kkv is detected in a punctate pattern throughout stages 16–17 and not in lines or bundles, in contrast to wild-type embryos. In all panels anterior is to the left and scale bars = 10 μm.

The following figure supplement is available for figure 3:

**Figure supplement 1.** Kkv co-localises with F-actin and chitin rings at embryonic and larval stages.

actin fibers. In *tal/pri* mutants, actin fiber organisation is disrupted and taenidia are not properly formed, reinforcing the idea of a functional role for Kkv distribution.

## Chitin deposition and actin bundling contribute to proper taenidial fold organisation

We next examined the contribution of chitin deposition to the organisation of taenidia. As when studying the contribution of tracheal actin rings to this process, we chose mutants that do not completely inhibit chitin deposition, as these mutants would probably heavily impair tracheal development, thus hindering specific analysis of the morphogenesis of taenidia. Thus, we turned to

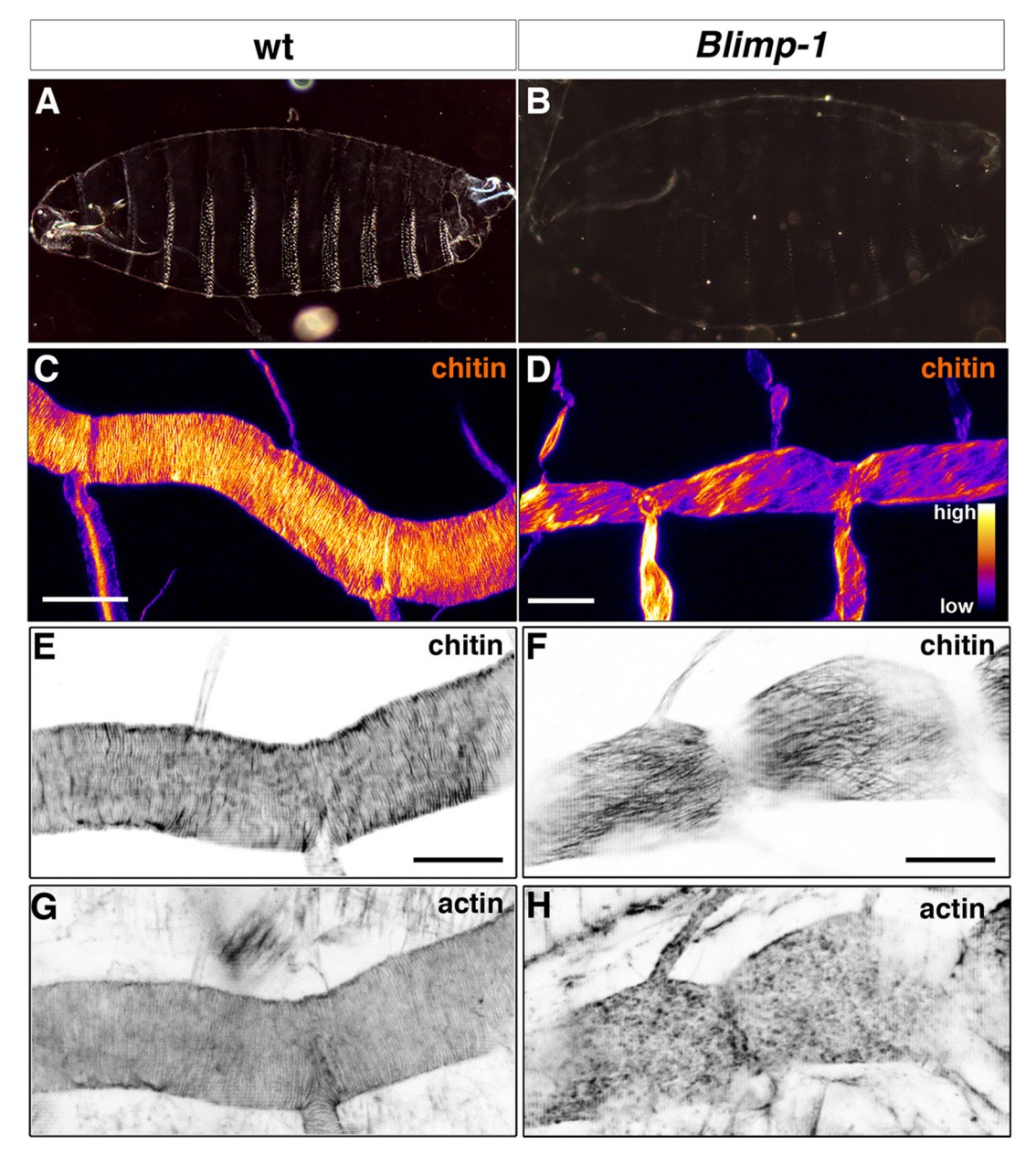

**Figure 4.** Taenidial folds, F-actin bundles, and chitin levels in *Blimp-1* mutant embryos. (A-B) Cuticle preparations of wild-type (A) and *Blimp-1* (B) embryos visualised under dark field. In the *Blimp-1* mutant embryo, the cuticle and the denticle belts, chitin structures at the epidermis, are faint when compared to wild-type preparation. (C-D) Stage 17 *Blimp-1* heterozygous (control, C) and *Blimp-1* homozygous mutant (D) embryos stained with fluostain to label chitin structures. After acquisition in the same conditions, the images are converted into colour-coded LUTs in which different levels of fluorescent signals are matched with different colours. The colour code is shown on the lower right hand of panel D. While in the control DT mostly red and yellow stains are observed, in the *Blimp-1* mutant DT there are mostly purple and red stains, indicating that the fluorescent signal level of fluostain is lower in the *Blimp-1* mutant DT. Scale bars = 10 μm. (E-H) Wild-type (E, G) and *Blimp-1* mutant embryos (F, H) stained with fluostain (E, F) to label taenidial folds and phalloidin (G, H) to label F-actin bundles. Both the taenidial folds and F-actin bundles run perpendicular to the tube axis in wild-type embryos (E, G) while in most of the *Blimp-1* mutants F-actin bundles fail to form (H) and taenidial folds run parallel to the tube axis (F). The images are single stacks of confocal sections. Scale bars = 10 μm.

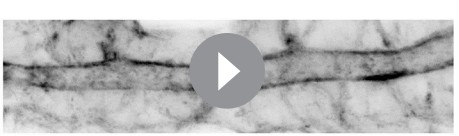

**Video 1.** Time-lapse images of a wild-type embryo carrying *btl::MoeGFP* construct visualised from a lateral view. Note the accumulation of F-actin at the apical surface leading to the polymerisation of F-actin bundles in the form of rings at the end of the movie.

*Blimp-1*, an ecdysone response gene (*Chavoshi et al., 2010*; *Beckstead et al., 2005*) that encodes the *Drosophila* homolog of the transcriptional factor *B-lymphocyte-inducing maturation protein* gene and whose mutants have been reported to have misshapen trachea almost completely devoid of taenidia (*Ng et al., 2006*).

Indeed, *Blimp-1* mutant embryos were grossly inflated compared to the wild-type (*Figure 4A, B*), a phenotype associated with weaker embryonic cuticles caused by mutations impairing the deposition or organisation of chitin (*Ostrowski et al., 2002*). Consistent with this observation, *Blimp-1* mutants showed a pale ectodermal cuticle with smaller denticles (*Figure 4A, B*), although their phenotype is weaker than that of the *kkv* chitin synthase mutants (*Ostrowski et al., 2002*). This observation suggests that, while chitin deposition is severely impaired, some still accumulated in the cuticle of *Blimp-1* mutant embryos. In support of this hypothesis, we detected lower levels of fluostain signal in the trachea of *Blimp-1* mutants compared to the wild-type (*Figure 4C, D*). Thus, we expected to find similarly less conspicuous taenidia, which was indeed the case. However, the most obvious abnormal feature of taenidia was their pattern, as they were not organised in folds perpendicular to the tube axis but instead ran parallel to it (*Figure 4F*). Given the close correlation between taenidia and actin bundle organisation, we examined actin arrangement in *Blimp-1* mutants and found that it was severely impaired. In most *Blimp-1* mutants examined (67%, n = 18), we did not observe tracheal actin rings (*Figure 4H*, *Videos 1* and *2*). However, in the mutant embryos in which we detected apical actin bundles (33%, n = 18), these were oriented in parallel to the tube length (data not shown) like the chitin structures (*Figure 4F*). Thus, as is the case for the other mutant genotypes examined so far, in *Blimp-1* embryos the lack of a proper arrangement of taenidial folds correlates with either the absence or abnormal pattern of actin rings.

Detailed ultrastructural analysis by TEM confirmed the close interplay between actin and chitin in both *tal/pri* and *Blimp-1* mutants. In wild-type embryos, each taenidium is formed by a plasma membrane protrusion and the taenidia have a regular shape (*Figure 5A, D*). Arrangement of plasma membrane protrusions in *tal/pri* and *Blimp-1* mutant tracheal cells is irregular (*Figure 5B, C, E, F*). At the end of embryogenesis, whereas the breadth of these taenidia is very constant in wt animals, it is highly variable in *tal/pri* and *Blimp-1* mutants (wt 243 nm +/-8%, n = 20; *pri* 346 nm, +/-35%; n = 20 and *Blimp-1* 543 nm +/- 45%, n =13, measured at the basis of the taenidia contacting the plasma membrane, see 'Materials and methods' for details). This result is in line with the finding that proper F-actin ring organisation and chitin deposition are necessary for taenidial morphogenesis.

## The chitinous aECM ensures the organisation of the tracheal actin rings

The observation of an effect of a mutation in a gene required for proper chitin arrangement on actin bundling was unexpected. To assess whether the effect of *Blimp-1* mutations on actin organisation was indeed a consequence of abnormal chitin deposition in the tracheal cuticle rather than the result of a direct and yet unknown role of *Blimp-1* in F-actin bundling, we examined tracheal actin organisation in mutants for *kkv*, a gene required for chitin morphogenesis only (*Moussian et al., 2005*). Surprisingly, *kkv* mutants also lacked actin rings (*Figure 6D*), thereby indicating a feedback role of proper chitin-mediated tracheal cuticle in F-actin organisation. In addition, F-actin bundles formed normally and thereafter collapsed in *kkv* mutants (*Figure 6I-L*, *Videos 3* and *4*). This finding indicates that a proper cuticle is not required for the establishment of the F-actin rings but instead for their maintenance. This implies that proper chitin deposition/

**Video 2.** Time-lapse images of a *Blimp-1* mutant embryo carrying *btl::MoeGFP* construct visualised from a lateral view. Note that the accumulation of F-actin at the apical surface occurs but F-actin fail to form bundles in the form of rings at the end of the movie (except at the fusion points).

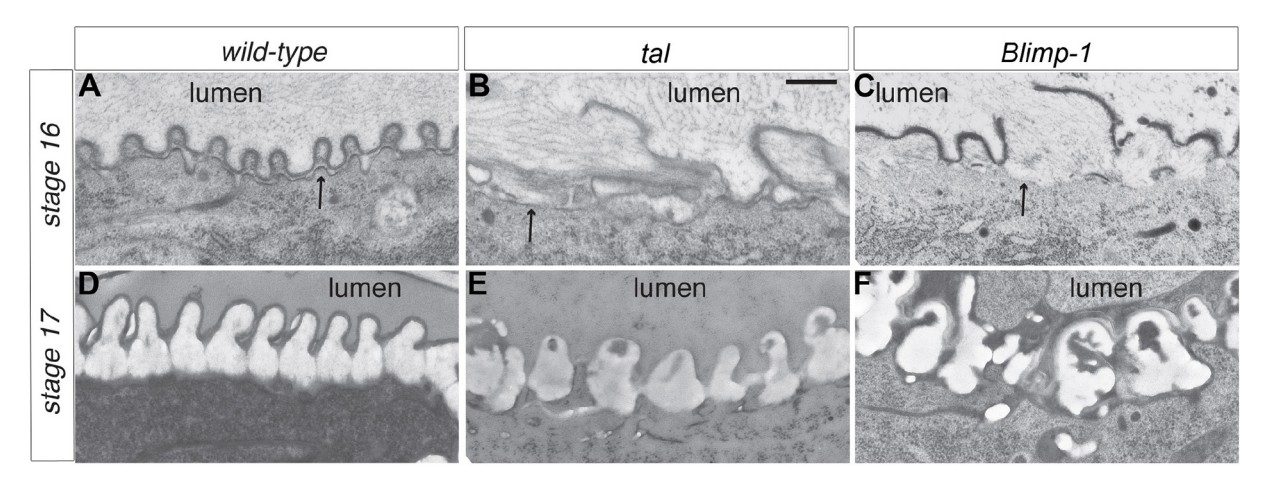

**Figure 5.** Taenidial structure in *tal/pri* and *Blimp-1* embryos. (A-F) Electron-micrographs of longitudinal sections of the dorsal trunk in wild-type, *tal/pri* and *Blimp-1* embryos. (**A**) In wild-type stage 16 embryos, regular protrusions of the apical plasma membrane (arrow) of dorsal trunk cells produce the extracellular taenidial cuticle that mainly consists of the inner procuticle (pro) and the outer envelope (env). (**B,C**) In *tal/pri* and *Blimp-1* mutant stage 16 embryos, protrusions of the apical plasma membrane are irregular with occasionally extended flat regions. The taenidial cuticle follows these irregularities, and the envelope is discontinuous. (**D**) In wild-type stage 17 embryos, prior to hatching, the fully differentiated taenidial cuticle is characterised by folds of nearly equal breadth. (**E,F**) In contrast, in *tal/pri* and *Blimp-1* mutant stage 17 embryos the breadth of the differentiated taenidial folds is highly variable. Scale bar 500 nm applies to all electron-micrographs.

organisation contributes to ensure the proper organisation and stability of the apical F-actin rings.

## Taenidial organisation and cell shape

How could the apical chitin in the ECM influence actin bundling? We observed that both *kkv* and *Blimp-1* mutations had an effect on tracheal cell shape. In the wild-type trachea, the cells of the DT were organised such that the longest axis of their apical shape is parallel to the tube axis. However, in both *Blimp-1* and *kkv* mutant trachea (*Figure 7B, C*), the anteroposterior elongation of the cells of the DT was lost, causing cells to be more square shaped. Thus, we hypothesised that the change in taenidial orientation in *kkv* and *Blimp-1* mutants could be attributed to the alteration in the overall orientation or shape of the tracheal cells. Interestingly, a modification of cell shape/orientation also occurs in embryos mutant for the Src-family kinase Src42A (*Förster and Luschnig, 2012*). However, and as previously reported for F-actin (*Förster and Luschnig, 2012*), we found taenidia to follow the same organisation in *Src42A* mutant embryos as the wild-type (*Figure 7E*) indicating that proper organisation of taenidia can be uncoupled from correct tracheal cell shape/orientation and thus that the former is not merely a consequence of the latter.

## Individual cells contribute to the supracellular organisation of taenidia

Having identified and characterised genes that specifically affect taenidial patterning, we examined the individual cell contributions to this supracellular organisation by impairing genetic functions in mosaics. We were unable to generate mosaics by mitotic recombination since there are no cell divisions after tracheal invagination and RNAi-mediated knockdown often does not work in *Drosophila* embryogenesis; this was indeed the case upon expression of UAS-RNAi constructs for either *tal/pri* or *Blimp-1* in the embryonic tracheal cells. Thus, we turned to alternative approaches to produce tracheal cellular chimeras.

First, we took advantage of the effect of *Blimp-1* overexpression on taenidial formation (*Ng et al., 2006*). To generate tracheal DTs with distinct cellular composition, we used an *AbdB-Gal4* line that drives expression only in the posterior part of the embryo (*Förster and Luschnig, 2012*; *Förster et al., 2010*). This approach also allowed us to have an internal control within the same embryo. Upon expression of *UASBlimp-1* under these conditions, lower levels of chitin were

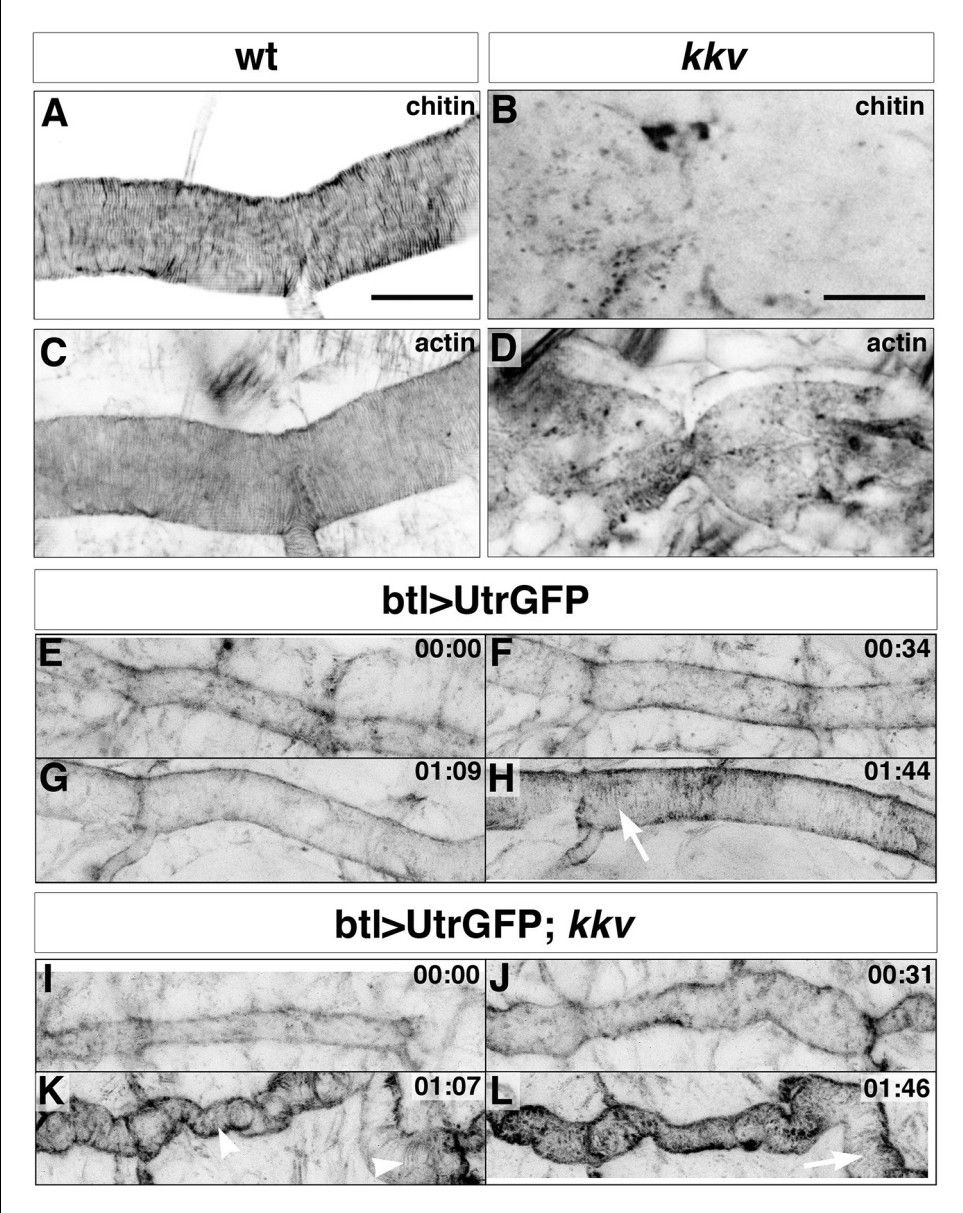

**Figure 6.** Taenidial folds and F-actin bundles in *kkv* mutant embryos. (**A-D**) *Wild-type* (**A, C**) and *kkv* mutant (**B, D**) mutant embryos stained with fluostain to label taenidial folds (**A, B**) and phalloidin to label F-actin bundles (**C, D**). The taenidial folds and F-actin bundles run perpendicular to the tube axis in the *wild-type* embryo (**A, C**) while in *kkv* mutant embryos taenidial folds are absent and F-actin bundles fail to form (**B-D**). The images are single stacks of confocal sections. Scale bars = 10 μm. (**E-L**) Time-lapse images of *wild-type* (**E-H**) and *kkv* mutant (**I-L**) embryos carrying *btlGAL4UASUtrGFP* to visualise actin in live embryos. In the *wild-type* embryo, F-actin bundles (arrow) become visible at the end of time-lapse imaging (**H**) while in the *kkv* mutant F-actin bundles form transiently (**K**, arrowhead) and then disappear (**L**, arrow).

detected in the posterior metameres (*Figure 8A*). Thus, chitin deposition seems to be highly dependent on the levels of Blimp-1 activity as both loss-of-function mutations and overexpression of *Blimp-1* induce low levels of chitin. We also noted that overexpression of *Blimp-1* gives rise to tracheal cells with a less elongated apical side (*Figure 8A''*), like that of *Blimp-1* and *kkv* mutants (*Figure 7B, C*). We then examined the trachea at the border of the *AbdB-Gal4* domain, finding a perfect correlation between the different physical appearance of taenidia and cells and their

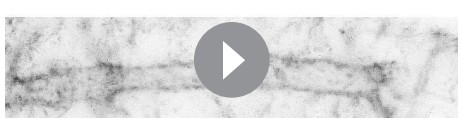

**Video 3.** Time-lapse images of a wild-type embryo carrying *btl>UtrGFP* visualised from a lateral view. Note the accumulation of F-actin at the apical surface leading to the polymerisation of F-actin bundles in the form of rings at the end of the movie.

genotype, with either wild-type or increased levels of *Blimp-1* (*Figure 8A*). We then generated flip-out clones expressing *Blimp-1* in a *wild-type* background and obtained similar results in these clones (*Figure 8B, C*, arrows). Thus, we conclude that *Blimp-1* regulates chitin accumulation in a cell-autonomous manner and that each cell contributes independently to the chitin deposition of their corresponding segments of the taenidial folds.

As a second approach to mosaic analysis, we used the same *AbdBGal4* line to drive expression of *tal/pri* and *Blimp-1* in *tal/pri* and in *Blimp-1* loss-of-function mutant backgrounds, respectively. For both mutants, we saw a rescuing effect in the posterior tracheal metameres as taenidial folds became organised perpendicularly to the tube length (*Figure 8D, E*). Using this approach, we were able to generate borders of cells with and without *tal/pri* and *Blimp-1* function and analyse taenidia in these conditions (red dotted line, *Figure 8D, E*). In the case of the *tal/pri* rescue experiment, we detected a difference between the cells expressing the wild-type *tal/pri* gene and those with a wild-type phenotype, an observation consistent with the non-cell autonomous function of the Tal/Pri peptides (*Kondo et al., 2007*; *Galindo et al., 2007*; *Chanut-Delalande et al., 2014*). However, in the case of the *Blimp-1* rescue experiment, taenidia tended to follow the orientation dictated by the genotype of their respective cells (*Figure 8D and F*). Moreover, and due to the expression domain of the *AbdBGAL4* driver not being completely continuous, we observed single cells of one of the genotypes surrounded by cells of the other and could detect either mutant cells with a longitudinal arrangement of the taenidia (*Figure 8F'*, arrow) or 'rescued' cells with a perpendicular arrangement (*Figure 8F'*, arrowhead); in this case, there was a correlation between the physical appearance of taenidia and the corresponding cell genotype. Interestingly, we also detected intermediate orientations between the prototypical longitudinal taenidia in the mutant domain and the perpendicular ones in the rescued domain (*Figure 8F'*, asterisk). These results suggest that cells 'adapt' the orientation of 'their' segments of the taenidia to the global orientation of the segments of the taenidia contributed by neighbouring cells.

These results show that tracheal taenidia can form proper rings even when the neighbouring cells do not. This indicates that, to a certain degree, segments of taenidia can organise properly even in the absence of proper subjacent actin rings provided that the segments of taenidia contributed by the neighbouring cells are properly organised.

## The role of cellular junctions in the supracellular organisation of taenidia

The pattern of the tracheal actin rings prefigures that of taenidial folds, and the supracellular organisation of taenidia is a consequence of the shared orientation of actin bundles between neighbouring cells. A communal orientation of the intracellular apical actin bundles could arise from cells responding to a common extracellular cue. Alternatively, although not mutually exclusive, tracheal cells could coordinate the orientation of their apical actin bundles by means of their cell-cell junctions. Therefore, we sought to analyse the role of cellular junctions in the supracellular organisation of taenidial folds. We first attempted to downregulate cell-junction components in the embryonic trachea by means of expressing their corresponding double-stranded RNA constructs; however, we did not obtain a clear taenidial mutant phenotype. As mentioned before, RNAi-mediated knockdown often does not work during *Drosophila* embryogenesis and so we repeated the same experiments in the trachea of the 3rd larval instar. Upon expression of a dsRNA targeted at the gene encoding alpha-catenin (α-cat), we

**Video 4.** Time-lapse images of a *kkv* mutant embryo carrying *btl>UtrGFP* visualised from a lateral view. Note the accumulation of F-actin at the apical surface leading to the polymerisation of transient F-actin bundles in the form of rings that dissociate at the end of the movie.

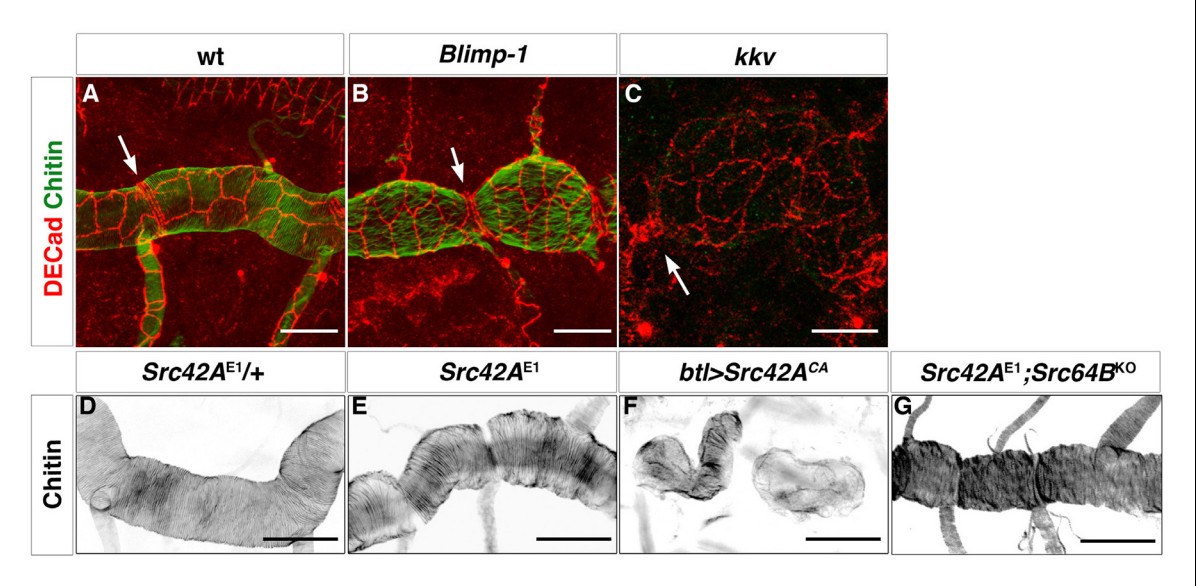

**Figure 7.** Apical cell shape in the *Blimp-1* and *kkv* mutant embryos. *Wild-type* (**A**), *Blimp-1* mutant (**B**) and *kkv* mutant (**C**) embryos stained with anti-DE-cad (red) to label the cell surface and fluostain (green) to label taenidial folds. In the *Blimp-1* mutant embryo, apical cell shape is elongated perpendicular to the tube axis (**B**), whereas it is mostly parallel to the tube axis in the *wild-type* embryo (**A**). In the *kkv* mutant embryo, apical cell shape resembles more the *Blimp-1* phenotype (**C**). The distinct apical cell shape of the fusion cells (**A**, arrow) does not seem to be affected by the loss of function of *Blimp-1* (**B**, arrow) but is affected by the absence of *kkv* (**C**, arrow). The images are projections of confocal sections. Scale bars = 10 μm. (**D–F**) *Wild-type* (**D**) and *Src42A* mutant (**E**) embryos and the effect of constitutively activated Src42A on tracheal cells (**F**) of embryos stained with fluostain to label taenidial folds. The taenidial folds run perpendicular to the tube axis in *wild-type* (**D**), *Src42A* mutant (**E**) and *Src42A;Src64B* double mutant (**G**) embryos, whereas they are not properly formed in overexpression of Src42A^CA embryos (**F**). The images are projections of confocal sections. Scale bars =10 μm.

found that DE-cad was strongly downregulated (undetected by antibody staining), demonstrating that adherens junction (AJ) formation is disrupted in the DT of these embryos (*Figure 9—figure supplement 1H*). By downregulating AJs, we found that in these embryos taenidia organised in groups of independent units, not encompassing the overall tracheal tube diameter (*Figure 9 B*) (cell-cell junction taenidial disruption was detected in 123 out of 156 DT cells analysed (79%), n = 18 larvae). Strikingly, despite the observed role of cell-cell junctions in taenidial continuity, these are still placed perpendicular to the tube length. This suggests that tracheal cells are able to sense a global orientation cue and align their actin bundles appropriately (*Matusek et al., 2006*), despite having their continuity disrupted at the level of the cell-cell junctions. Also, in this mutant background, taenidia organisation mimicked the distribution of actin bundles (*Figure 9—figure supplement 1*), which were similarly restricted to these independent units (100%, n = 17 larvae). Double labelling these trachea with fluostain and an apical cell membrane marker showed that these units corresponded to single cells (*Figure 9B and C*), thereby confirming the role of cell-junctions in coordinating the pattern of F-actin bundles from neighbouring tracheal cells and ensuring a supracellular taenidia organisation encompassing the overall tracheal tube.

## The chitin apical ECM influences the levels of Src42A phosphorylation

Actin bundles from tracheal cells might organise the chitin ECM by influencing the distribution of the Kkv chitin synthase. Less clear is how the chitin ECM could ensure the organisation of the tracheal actin rings. Previous work has also reported other effects of chitin on cytoskeletal organisation (*Tonning et al., 2005*), and in this case neither is it known how this effect is mediated. To advance in this direction, we sought to identify the proteins that could act as a link between the extracellular environment and the cytoskeletal organisation. To this end, we focused our attention on Src42A because of the role of the Src kinases in mediating extracellular signals to modulate actin

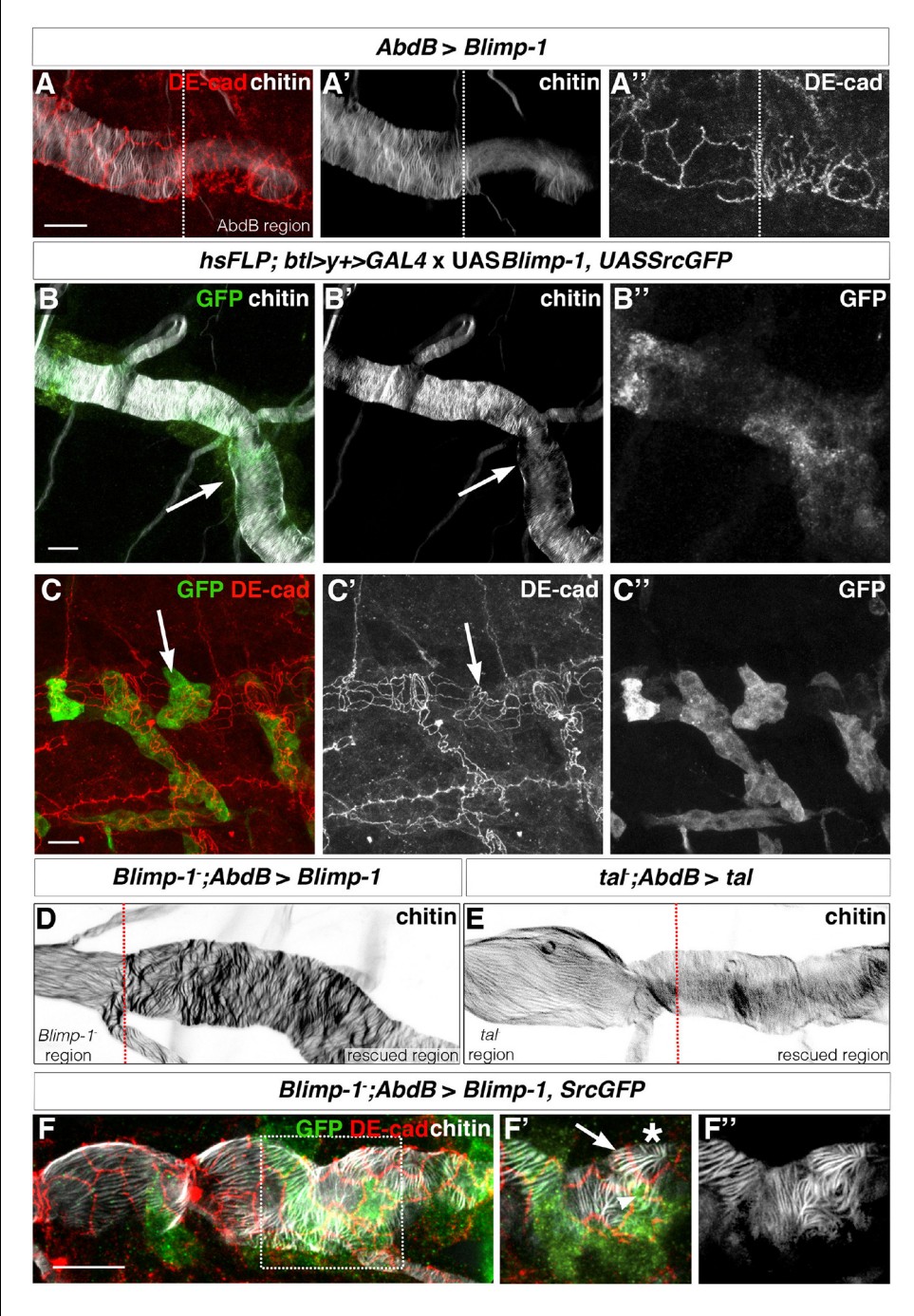

**Figure 8.** The orientation of taenidial folds is not regulated autonomously. (**A**) *AbdBGAL4*-driven *UASBlimp-1* expression in a stage 17 wild-type embryo that is stained with fluostain to label taenidial folds (**A, A'**) and DE-cad to label adherens junctions (**A, A''**). In the region where *Blimp-1* is overexpressed, chitin is reduced and the apical surfaces of cells are altered. (**B**) An embryo with flip-out clones of overexpression of *Blimp-1* stained with fluostain (grey) to label taenidial folds and anti-GFP (green) to detect expression of *UASSrcGFP* construct (and hence the clones). Note that the taenidial folds that are formed within the clones (arrow) are thinner than neighbouring taenidial folds formed by the cells outside the clones. The co-stainings are shown in the merge image (**B**). The chitin and anti-GFP stainings are shown separately (**B', B''**). (**C**) Embryos with flip-out clones of overexpression of *Blimp-1* stained with anti-DE-cad (red) to label the apical surface of the cells and anti-GFP (green) to detect expression of *UASSrcGFP* construct (hence the clones). Note that the shape of the apical surface of the cells in the clones (arrow) is altered when compared to the cells outside the clones. The co-stainings are shown in the merge

*Figure 8 continued on next page*

*Figure 8 continued*

image (**C**). The anti-DE-cad and anti-GFP stainings are shown separately (**C'**, **C''**). (**D, E**) *Blimp-1* mutant (**D**) and *tal/pri* mutant (**E**) embryos, carrying *AbdB-GAL4UASBlimp-1* (**D**) and *AbdBGAL4UASpri* (**E**) constructs, respectively, stained with fluostain to label taenidial folds (**D, E**). The orientation of taenidial folds that run parallel to the tube axis in *Blimp-1* and *tal/pri* mutant regions are rescued at the posterior part of the embryos, the rescued regions. The red dotted line separates the mutant from the rescued region, according to expression of GFP in the AbdB domain. (**F**) *Blimp-1* mutant embryo, carrying *UASBlimp-1* and *AbdBGAL4* constructs, stained with fluostain (grey) to label taenidial folds, GFP to show the AbdB domain and anti-DE-cad (red) to label apical surface of the cells. The *UASBlimp-1* construct is in a chromosome also carrying the *UASSrcGFP* construct and thus the anti-GFP (green) marks the area of *UASBlimp-1* expression. **F'** is a magnification of the rectangular region marked in **F**. The taenidial folds in the cell marked with an arrow (**F'**) run parallel to the tube axis and are not 'rescued'. In contrast, the taenidial folds in the cell marked with an arrowhead (**F'**) run perpendicular to the tube axis, because this cell carries the *UASBlimp-1* construct and hence it is 'rescued'. The cell marked with an asterisk has 'intermediate' taenidia displaying parallel and perpendicular orientation, dependent on the neighbouring cells. The triple-stainings are shown in the merge images (**F, F'**). The fluostain staining is shown separately (**F''**) to better assess the taenidial fold orientation. Scale bars 10 μm.

organisation (for a review see [*Parsons and Parsons, 2004*]) and because of the influence of constitutive Src42A activity in taenidial organisation. In particular, constitutively active mutations of Src42A disturb tracheal actin bundling (*Förster and Luschnig, 2012*) and, consequently, taenidial formation (*Figure 7 F*).

We observed higher levels of Src42A phosphorylation in *kkv* and *Blimp-1* mutants (data not shown). In the trachea, these increased levels have been shown to be coupled to an enhanced Src42A activity (*Förster and Luschnig, 2012*; *Nelson et al., 2012*). However, the differences in Src42A phosphorylation were too variable to unequivocally analyse differences in levels. To clearly ascertain this effect, we used the *AbdBGal4* line, which allowed us to generate internal controls within the same embryos. Although the *kkv* RNAi inactivation appeared to be highly inefficient in this context as we obtained very few embryos with lower levels of chitin at their posterior tracheal metameres, in all these cases (n = 4) there was an increased level of phosphorylated Src42A in the metameres where *kkv* expression was downregulated (*Figure 10A, D*). This observation was confirmed by the difference in levels of phosphorylated Src42A between *kkv* mutant and *kkv*-rescued cells (*Figure 10C, F*). In addition, lower levels of phosphorylated Src42A were detected in regions with higher chitin levels upon Blimp-1 rescue of the posterior tracheal region of *Blimp-1* mutant embryos (*Figure 10B, E*). This observation was confirmed by Kkv rescue of *kkv* embryos (*Figure 10C, F*). To check whether these differences could be attributed to differences in overall Src protein levels, we analysed Src protein in *kkv* mutant embryos and in embryos where Kkv was downregulated at the posterior end. In all cases, we could not detect any differences in Src protein levels in *kkv* mutant cells (*Figure 10—figure supplement 1A–D*). Furthermore, we checked for the anti-pSrc (pY418) specificity in *Src42A* mutants and confirmed that there is no cross-reactivity with other phosphorylated proteins in the embryo (*Figure 10—figure supplement 1A–D*) (*Förster and Luschnig, 2012*; *Shindo et al., 2008*).

Thus, given that downregulation of chitin synthesis leads to an increased phosphorylation of Src42A (*Figure 9A-C*) and that an increased activity of Src42A disturbs actin bundling (*Förster and Luschnig, 2012*), we propose that Src42A is one of the mediators of the extracellular chitin matrix in ensuring actin bundle organisation.

Of note and as mentioned above, while constitutive non-regulated activity of Src42A is sufficient to disturb taenidia, taenidia are normally organised in *Src42A* loss-of-function mutants indicating that the wild-type function of Src42A is not an absolute requirement for the wild-type taenidial patterning. A functional redundancy could account for this observation and Src64 would be a likely candidate for such a phenomenon. However, we did not detect any actin ring phenotype in the trachea of double mutant embryos for both *Src42A* and *Src64B* (*Figure 7G*). Thus, and irrespective of the nature of the role of Src42A activity on actin ring patterning, these results suggest that the feedback mechanism of the chitin aECM through Src42A is not mainly in instructing actin ring organisation but in preventing its disturbance by downregulating Src42A phosphorylation to levels compatible with

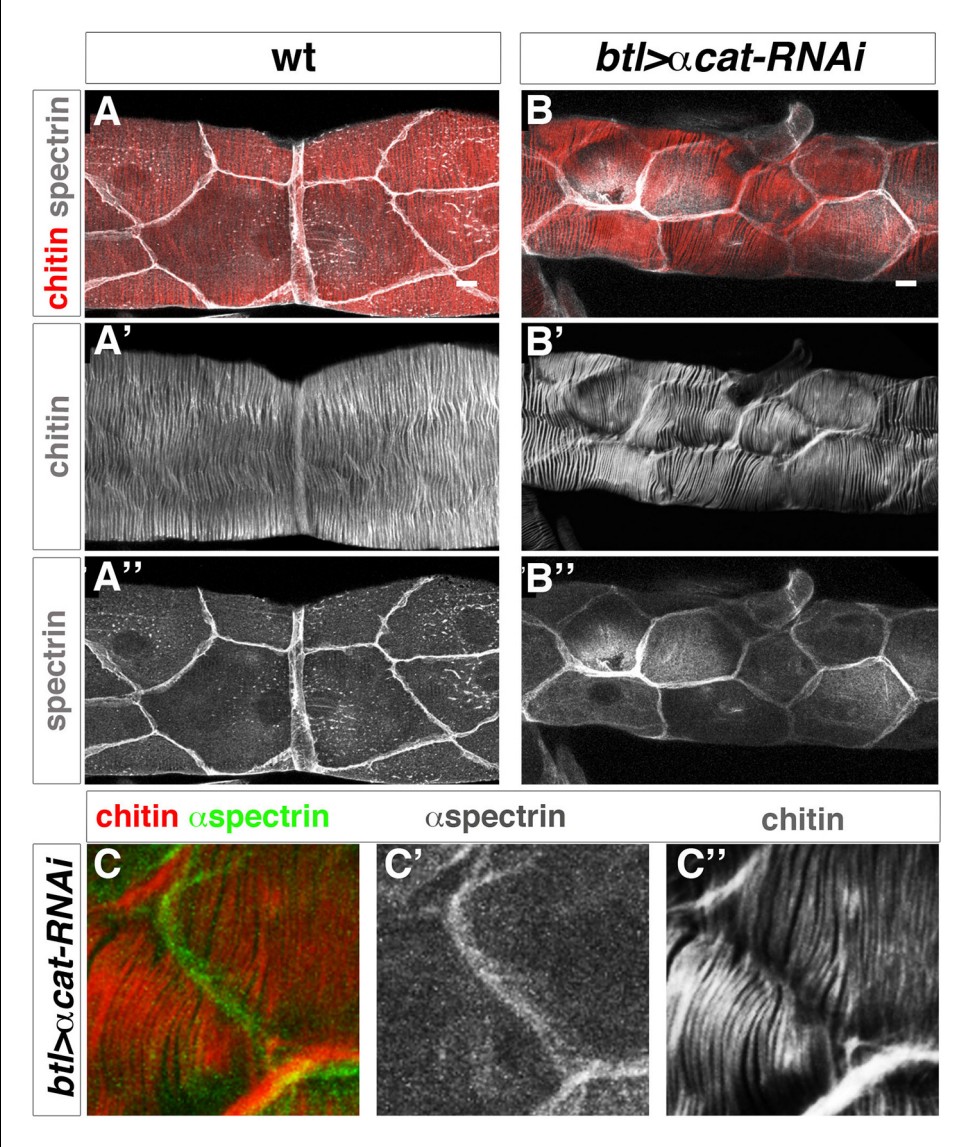

**Figure 9.** Taenidial folds in tracheal tubes with impaired cellular junctions. (A, B, C) Wild-type 3rd instar larval tracheas, carrying either no constructs (A) or *btl-GAL4* and *α-cat-RNAi* constructs (B, C) stained with fluostain (red or grey) to label taenidial folds and anti-Spec (grey in A, B and C' and green in C) to label apical cell boundaries. The continuous taenidial folds in the wild-type larval trachea (A) become discontinuous at the apical cell boundaries upon down regulation of cellular junction components (B, C). The co-stainings are shown in the merge images (A, B and C). The fluostain (A', B', C'') and anti-Spec (A'', B'' and C') stainings are shown separately. Scale bars 10 μm.

The following figure supplement is available for figure 9:

**Figure supplement 1.** Taenidial folds in tracheal tubes with impaired cellular junctions.

proper actin ring organisation. Accordingly, genetic removal of *Src42A* in *Blimp-1* homozygous embryos, rescues the taenidial organisation phenotype to wild-type levels, while keeping other *Blimp-1* mutant features in all double mutant embryos observed (n = 12, *Figure 10G-I*), confirming the biological role of Src42A in the mechanism underlying the supracellular organisation of the tracheal aECM.

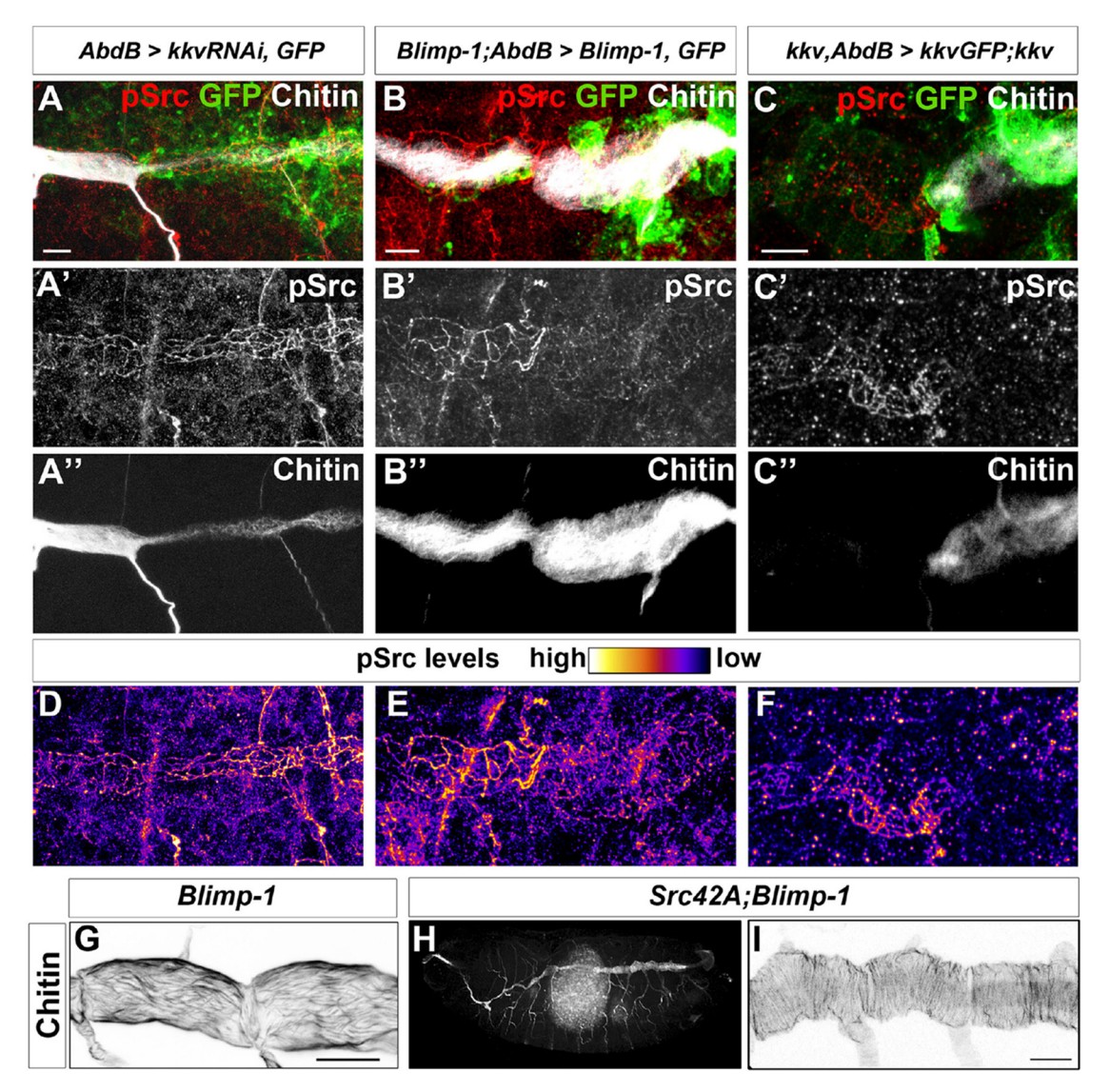

**Figure 10.** Chitin levels modulate Src activation. (A and D) Wild-type embryo expressing *AbdBGAL4UASkkvRNAi* and *UASSrcGFP*, stained with anti-GFP (green, A) to visualise the AbdB domain, anti-pSrc (red, A') to detect activated Src42A, and fluostain (grey, A'') to label chitin. Levels of pSrc are better visualised in D, with a different colour-coded LUT. (B and E) *Blimp-1* mutant embryo, expressing *AbdBGAL4UASBlimp-1* and *UASSrcGFP*, to rescue the *Blimp-1* phenotype at the posterior end of the embryo, stained with anti-GFP (green, B) to visualise the AbdB domain, anti-pSrc (red, B') to detect activated Src42A, and fluostain (grey, B'') to label chitin. Levels of pSrc are better appreciated in E, with a different colour-coded LUT. (C and F) *kkv* mutant embryo, expressing *AbdB-GAL4* and *UASkkvGFP*, to rescue the *kkv* phenotype at the posterior end of the embryo, stained with anti-GFP (green, C) to visualise the AbdB domain, anti-pSrc (red, C') to detect activated Src42A and fluostain (grey, C'') to label chitin. Levels of pSrc are better appreciated in F, with a different colour-coded LUT. Scale bars 10 μm. (G-I) Late stage 16 embryos stained with fluostain to visualise taenidial organisation. (G) Detail of *Blimp-1* dorsal trunk showing taenidial disorganisation; (H) whole *Src42A;Blimp-1* double mutant, showing some of the characteristic Blimp-1 tracheal phenotypes; (I) Detail of *Src42A;Blimp-1* double mutant dorsal trunk showing rescued taenidial organisation. Scale bars 25 μm.

The following figure supplement is available for figure 10:

**Figure supplement 1.** Src protein levels in *kkv* mutant trachea.

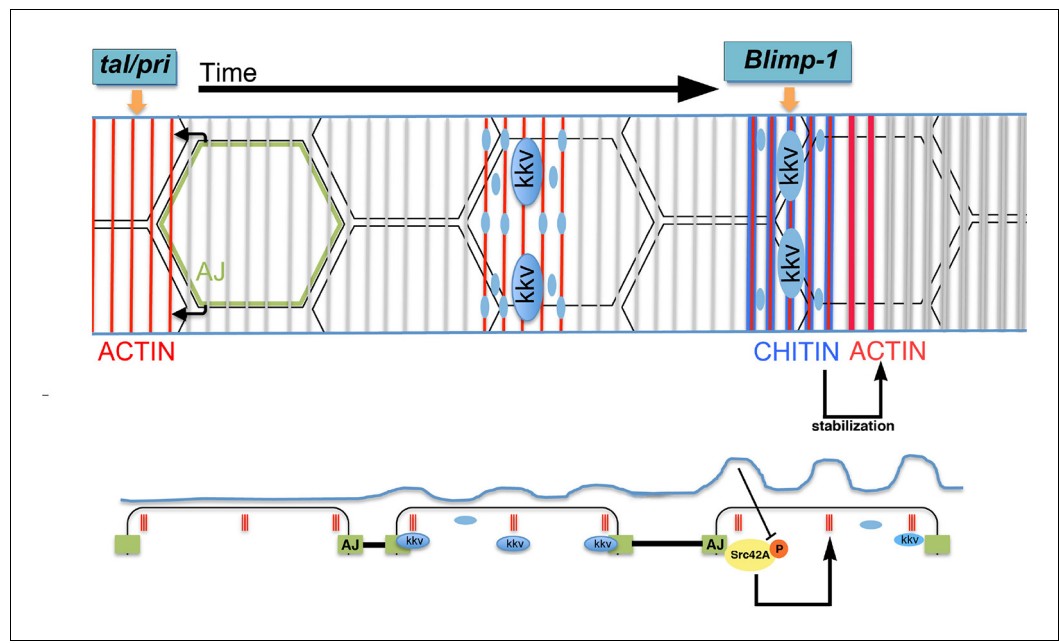

**Figure 11.** Proposed model for the generation of the supracellular taenidia. Model depicting the relationship between chitin synthesis and taenidial fold and F-actin organisation during taenidial fold formation. Drawn is the dorsal trunk at later stages of embryonic development and the effects of Tal/Pri, Blimp-1 and Kkv on F-actin and chitin organisation, respectively. This schematic representation depicts a time progression of events in taenidial formation. Large blue circles represent the Kkv localisation that is predominant within each taenidium, but it is also found in 'non-taenidial' regions of the membrane (smaller blue circles). The inhibitory effect of chitin in Src42A activation is depicted and well as the feedback from this inhibition back to the stabilisation of the F-actin rings.

## A model for the generation of the supracellular taenidia

The role for the apical chitin ECM in tracheal actin organisation indicates a feedback mechanism to generate the supracellular taenidial structures. In the light of the above and previously published results, we propose the following model for the formation of the taenidial folds that expand the overall diameter of the tracheal tube (*Figure 11*). On the one hand, actin polymerises in rings at the apical side of the tracheal cells in a *tal/pri*-dependent process; these actin rings are then required for the particular accumulation of the *kkv* chitin synthase and for the appearance of folds in the plasma membrane. In turn, *kkv* accumulation leads to a localised increased production and deposition of chitin along specific enriched stripes above the actin rings in a *Blimp-1*-mediated process. On the other hand, the cellular AJs are instrumental in ensuring that apical F-actin bundles from each cell follow a supracellular organ arrangement. It has to be noted that each cell appears to independently organise or maintain, to a certain degree, the proper orientation of their actin bundles, as determined by *Blimp-1* clonal analysis and the disruption of cell adhesion by downregulation of α-Cat and, consequently, DE-Cad. These results further suggest cell polarity along the circumferential axis of the tracheal tube. Nevertheless, this is not an absolute value as cells also have the capacity to modify the orientation of their sections of the taenidia to keep the continuity of these structures along the tube. In this regard, cell adhesion is central to ensure the continuity of the intracellular actin bundles as a patterning element for the overall tube. Subsequently, the chitin aECM feeds back on to the cellular architecture by stabilising F-actin bundling and cell shape via the modulation of Src42A phosphorylation levels. The combination of all these phenomena "*explain just how it is that the cells of the tracheal epithelium can cooperate unconsciously so as to form a helicoid [chitinous] thickening continuous from one end of the trachea to another*" (*Thompson, 1929*).

## Materials and methods

### *D. melanogaster* strains and genetics

All *D. melanogaster* strains were raised at 25°C under standard conditions. Mutant chromosomes were balanced over LacZ or GFP-labelled balancer chromosomes. Overexpression and rescue experiments were carried out either with *btl-GAL4* (kindly provided by M. Affolter) or *AbdB-GAL4* (kindly provided by E. Sánchez-Herrero) drivers at 22°C, 25°C or 29°C. *y1w118* (wild-type), *Blimp-1$^{KG09531}$*, *kkv$^1$*, *UASSrcGFP*, and *UASα-cat-RNAi* are described in FlyBase. *pri$^1$*, *pri$^2$*, *pri$^3$* and *btl::MoeGFP* (from S. Hayashi). UAS*Blimp-1* (this work); *UASUtrGFP* (from T. Lecuit), *UASStal$^{ε1}$* (from J. P. Couso), *UASkkvGFP* (from B. Moussian), *UASkkvRNAi* (VDRC). *hsFLP$^{122}$*; *btl::MoeRFP, btl >y$^+$ >GAL4* (from M. Affolter) and *UASSrc42A$^{CA}$* (from S. Luschnig).

### Immunohistochemistry, image acquisition, and processin

Standard protocols for immunostaining were applied. The following antibodies were used: rat anti-DE-cad (DCAD2, DSHB); rabbit anti-GFP(Molecular Probes, Eugene, OR); mAb2A12 (DSHB); mouse anti-Spec (DSHB); rabbit anti-pSrc pY418 (ThermoFisher); mouse anti-Src (Kojima and Saigo), and chicken anti-β-gal (Cappel). Biotinylated or Cy3-, Cy2- and Cy5-conjugated secondary antibodies (Jackson ImmunoResearch, West Grove, PA) were used at 1:250. For some fluorescent stainings, the signal was amplified using TSA (NEN Life Sciences , Boston, MA) when required. Chitin was visualised with Fluostain (Sigma) at 1 µg/ml or Chitin Binding Probe (CBP, our own, made according to NEB protocols). F-actin was visualised with Phalloidin (Sigma-Aldrich) at 1:50.

Confocal images of fixed embryos were obtained either with a Leica TCS-SPE, a Leica TCS-SP2, or a Leica TCS-SP5 system. Images were processed using Fiji and assembled using Photoshop.

### Cuticle preparation

Fully developed embryos were dechorionated in bleach, devitellinised by shaking in 100% methanol, and incubated over night at 65°C in Hoyer's medium mixed with lactic acid (1:1). Embryos were analysed by light microscopy using a Nikon Eclipse 80i microscope.

### Time-lapse imaging

Dechorionated embryos were immobilised with glue on a coverslip and covered with Oil 10-S Voltalef (VWR). To visualise in vivo F-actin bundling/ring formation, *btl::MoeGFP* and *UASUtrGFP* constructs were used in the indicated backgrounds. F-actin dynamics were imaged with a spectral confocal microscope Leica TCS SP5. The images were acquired every 5 min over Z stacks from stage 14–17 embryos for 2–3 hr. The movies were assembled using Fiji.

### Transmission electron microscopy

For ultrastructural analyses by TEM, wild-type and *Blimp-1* and *tal/pri* mutant embryos were immobilised by high-pressure freezing, fixed by freeze substitution, embedded in Epon, and sectioned as described previously (*Moussian et al., 2006b*). Images were taken on a CM10 electron microscope.

For quantification of taenidial breadth, the angles of the sections were in *Blimp-1* and tal/pri mutants as in the wild-type. In all cases, serial sections were examined to follow taenidial running direction. Measurements were done in the middle-most taenidia in all samples to avoid distortion effects at the edges of tangencial sections.

While this paper was being reviewed, it was reported that genetic depletion of the aECM caused dynamic movements of actin rings independently confirming our own observations (*Hannezo et al., 2015*). In addition, it was recently published that transient cell-junction anisotropies are on the basis of the actin ring and taenidia orientation (*Hosono et al., 2015*).

## Acknowledgements

We are grateful to M. Llimargas for critically reading the manuscript. We thank M Llimargas, E Sanchez-Herrero, T Kojima, JP Couso, S Luschnig, M Affolter, S Hayashi and H Ueda, the Developmental Studies Hybridoma Bank and the Bloomington Stock center for fly stocks and reagents. We thank L

Bardia, A Lladó and J Colombelli from the IRB-ADMF for assistance and advice with confocal microscopy and software and E Fuentes and N Martin for technical assistance.

## Additional information

### Funding

| Funder | Grant reference number | Author |
|---|---|---|
| Deutsche Forschungsgemeinschaft | MO1714/3 | Bernard Moussian |
| Ministerio de Economía y Competitividad | BFU2009-07629 | Arzu Öztürk-Çolak Sofia J Araujo Jordi Casanova |
| Ministerio de Economía y Competitividad | RYC-2007-00417 | Sofia J Araujo |

The funders had no role in study design, data collection and interpretation, or the decision to submit the work for publication.

### Author contributions

AÖÇ, BM, Acquisition of data, Analysis and interpretation of data, Drafting or revising the article; SJA, Conception and design, Acquisition of data, Analysis and interpretation of data, Drafting or revising the article; JC, Conception and design, Analysis and interpretation of data, Drafting or revising the article

### Author ORCIDs

Sofia J Araújo, http://orcid.org/0000-0002-4749-8913

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
