## [Decision Letter]

Thank you for submitting your work entitled "A feedback mechanism converts individual cell features into a supracellular ECM structure in *Drosophila* trachea" for peer review at *eLife*. Your submission has been favorably evaluated by K VijayRaghavan (Senior editor), a Reviewing editor, and two reviewers.

The reviewers have discussed the reviews with one another and the Reviewing editor has drafted this decision to help you prepare a revised submission.

Summary of the work:

The manuscript reports on an interesting interplay of the apical cytoskeleton and the extracellular matrix lining the tracheal tubes. The authors characterize the construction and alignment of parallel actin filaments and chitin ECM structures (taenidia) constructed perpendicular to the tube axis. They describe the phenotypes of tal/pri (peptides with both extracellular and intracellular functions), blimp (TF) mutants by in situ stainings and TEM. The new intriguing observation is that in blimp mutants the cytoskeletal filaments are disrupted but some remaining mis-oriented chitin assemblies are formed suggesting that the structure and alignment of the chitin polymers contributes to the construction of cytoskeletal elements. As src42 activity has been shown to control cell shape changes during tracheal tube growth the authors investigated the effect of blimp and kkv inactivation on the levels of phospho-src in the tubes. They show that p-src levels vary in response to the level/structure of intraluminal chitin. The paper closes with a model proposing the interplay between chitin ECM and the cytoskeleton indicating that blimp regulates chitin filament formation and tal/pri the construction and alignment of junctions.

Review summary:

The reviewers found the subject and the work interesting. They find that the authors have tackled an important problem (ECM – cell interactions) and that the paper is clearly written. The reviewers are also unanimous in feeling that the conclusions from the experiments are in places overstated, and the following revisions are needed for the paper to be accepted by *eLife*:

Major revisions:

1) Although the authors demonstrate that changes in ECM alter the levels of phospho-src, the absence of an actin ring phenotype in src mutants severely undercuts evidence for a feedback mechanism in matrix organization. The authors need to either establish a role for src42 in actin ring formation, for example by showing that src42; src64 double mutants have an actin ring phenotype at a stage when changes in phospho-src are observed, or by showing that ECM changes result in changes in phosphorylation or activity of Btk, which does have a ring phenotype.

2) Although the authors clearly show that ECM is involved in tracheal morphogenesis, the experiments in the manuscript do not, as stated in the Abstract, investigate the "stiffness" or other physical properties of the ECM. The authors need to either measure discussed physical properties or alter the text to accurately reflect the experimental data.

3) The perpendicular actin and its co-localization with kkvGFP are difficult to see in Figure 3. The authors need to provide more convincing, high-magnification pictures, with quantification, to support their claims. Similarly in Figure 4, the reported longitudinal chitin filaments are visible, but there are also some perpendicular arrangements, which undermine the argument that specific localization of kkv dictates ECM organization.

4) The authors need to be exact in their discussion of the a-cat mutant phenotype. Since a-cat is not an integral junctional adhesion protein, the presented data *do not* show the cell-cell junctions are key players ECM patterning. The authors could bolster the evidence for involvement of adherens junctions by examining E-cad mutants, or demonstrating that adherens junction formation is disrupted in a-cat mutants. Alternatively, the existing evidence could be discussed in terms of an apparent requirement for linkage actin cytoskeleton to adherens junction complexes. The authors should also address concerns that while a-cat mutants clearly have disrupted ECM patterning near cell junctions, there still appears to be some registration of the taenidial ring between cells in the a-cat mutants. The authors should comment on this, and consider alternative models for the importance of kkv localization.

5) While models are necessarily imprecise abstractions of data, the reviewers feel the presented model in Figure 11 over interprets multiple aspects of the presented data, including the colocalization of kkv with chitin, the importance of src in the feedback loop to actin ring organization and the specificity of tal and Blimp-1 acting only on actin and chitin. The authors should also check the signs of the arrows in the model, as well as the localization of src (which is shown as an extracellular protein), as there appears to be errors in the original Figure 11. It is also unclear that the figure shows a time series.

[Editors' note: further revisions were requested prior to acceptance, as described below.]

Thank you for re-submitting your work entitled "A feedback mechanism converts individual cell features into a supracellular ECM structure in *Drosophila* trachea" for further consideration at *eLife*. It is unusual for *eLife* to ask for a second revision, and in this case we agree that you have made many important changes, including to the model. Unfortunately, the central issue about the involvement of src has remained unanswered. For the manuscript to be considered for publication, please address the following two points. The first by performing one more set of experiments and the second by addressing the issue in the revised manuscript.

1) Requires new experiment

In the first review of the manuscript, we asked for evidence establishing a role for src42 in actin ring formation, for example by showing that src42; src64 double mutants have an actin ring phenotype at a stage when changes in phospho-src are observed, or by showing that ECM changes result in changes in phosphorylation or activity of Btk, which does have a ring phenotype. It is now clear that the src42; src64 double does not have an actin ring phenotype, and it is understandable that making a triple mutant for src42, src64 and btk is technically difficult and phospho-Btk reagents do not appear to be available. The new model in which feedback from the ECM to Src prevents Src over-activation that would disrupt actin ring formation is very elegant and does account for the known results, but an equally plausible possibility is that ECM-regulated changes in Src phosphorylation are "off pathway" and do not have biological significance. Consequently, it is unclear whether this manuscript has defined an important feedback loop between ECM and actin organization and thus whether the manuscript will have a substantial impact on the field.

For the paper to reach the standards of *eLife*, as stated in the original review, it is critical to establish a role Src in actin ring formation. If the current model is correct that Src is not required for actin ring formation, but that Src activity must not be too high, then it should be the case that a Blimp-1;src42 or kkv;src42 double will have intact actin rings because there is no Src activity to disrupt them. The reviewers appreciate that it is quite possible, perhaps even likely, that ECM will have a more complex role in actin ring formation than keeping Src activity low, and thus that the proposed double mutants will still have abnormal actin rings, but it is essential for this paper to demonstrate an actual role for Src in actin ring formation. The proposed double mutant experiments are not presented as required experiments, but rather as possible experiments that could, using loss-of-function approaches, demonstrate a role for Src in actin ring formation. Alternative experiments would be acceptable, as long as they demonstrate a biologically meaningful role for Src in actin ring formation.

Parenthetically, it is possible that the anti-p-src stainings in the paper reflect src activity. It is also possible that they also reflect p-Abl activity (M. Tamada et al., Dev. Cell, 2012, 309-319). Please include controls to show that the staining is eliminated in src mutants.

2) Modifications to the text

The last paragraph of the subsection “Tracheal actin rings relate to the spatial distribution of the chitin synthase” suggests kkv does not align with actin in tal/pri mutants. However, the actin patterns look strikingly similar to the taenidial patterns, and the last paragraph of the subsection “Chitin deposition and actin bundling contribute to proper taenidial fold organization “says there is "close interplay between actin and chitin in both tal/pri and Blimp-1 mutants". If kkv is not aligned with the actin in these mutants, this result actually undermines the argument that actin organizes kkv, which in turn produces an organized chitin pattern. Please clarify and explain in the text.

The issue raised above in #1 is critical for this manuscript to be accepted.

---

## [Author Response]

*1) Although the authors demonstrate that changes in ECM alter the levels of phospho-src, the absence of an actin ring phenotype in src mutants severely undercuts evidence for a feedback mechanism in matrix organization. The authors need to either establish a role for src42 in actin ring formation, for example by showing that src42; src64 double mutants have an actin ring phenotype at a stage when changes in phospho-src are observed, or by showing that ECM changes result in changes in phosphorylation or activity of Btk, which does have a ring phenotype.*

The reviewers are completely right on the fact that it is not clear at all which is the role of Src42 in actin organization. Indeed, as they point out, there is not an actin ring phenotype in *src42* mutants (Luschnig, Nat Cell Biol, 2012). However, in the same work it has been shown that constitutive phosphorylation of Src42 produces an alteration of the same actin structure. Thus, while Src42 protein is not an absolute requirement for actin ring formation (see below), its phosphorylation needs to be below a given threshold to ensure that the actin rings form properly.

And precisely, what our results show is that mutants affecting the chitin ECM produce an increase in Src42 phosphorylation. In other words, a proper ECM is required to "downregulate" Src42 phosphorylation and thus to maintain the actin ring organization. Thus, we think this points out to a mechanism by which matrix organization ensures a proper actin organization, which in turn is critical to ensure a proper ECM, therefore establishing what we call a feedback mechanism.

Having said so, the puzzle remains: high phosphorylation of Src42 disrupts actin ring organization while the protein itself is dispensable for actin ring organization (Forster and Luschnig, Nat Cell Biol, 2012 and Nelson et al., Nat Cell Biol, 2012). A possible explanation to account for this observation could be another protein having a redundant function; among those Src64 would be a likely candidate. In this regard and as suggested, we examined *Src42;Src64* mutant embryos but we did not observe any actin ring phenotype in these double mutants. This observation does not completely rule out the redundant hypothesis as other proteins, such as Btk also mentioned by the reviewers, could substitute for Src42 and/or Src64 functions on what regards actin ring organization. In this regard, we would like to mention that Btk is supposed to be downstream of both *Src42A* and *Src64B* and there are some data showing that Btk enhances the hypomorph phenotype of *Src42A* (Tateno et al. Science 200 and Roulier et al., Mol Cell 1998) while, on the other hand, Btk is also said to be activated in a Src-independent way by other kinds of kinases (Takahashi, et al., Development 2005; Matusek et al. Development 2006, Roulier et al. Mol Cell 1998, Lu et al., EMBO J. 2004 and Thomas and Wieschaus, Development 2004). Altogether, the analysis of a possible redundancy involving three or more elements becomes very complicated. On the other hand, we have not been able to identify in the literature an anti-p-Btk antibody which would have allowed us to assess changes in Btk phosphorylation upon changes in the ECM.

Nevertheless, whereas the exact role of these proteins in actin ring organization remains an open issue, what is clear is that an increased phosphorylation of Src42A has to be prevented to allow for a proper actin ring organization and our results precisely show that ECM chitin organization prevents an increased phosphorylation of Src42A, thus providing a causal link between proper ECM and proper actin ring organization. We have now rewritten the manuscript to include the new data and to discuss further this point addressing the comments from the referees.

*2) Although the authors clearly show that ECM is involved in tracheal morphogenesis, the experiments in the manuscript do not, as stated in the Abstract, investigate the "stiffness" or other physical properties of the ECM. The authors need to either measure discussed physical properties or alter the text to accurately reflect the experimental data.*

We have changed the text accordingly. More importantly, we have changed the reference to ECM stiffness in the Abstract.

*3) The perpendicular actin and its co-localization with kkvGFP are difficult to see in Figure 3. The authors need to provide more convincing, high-magnification pictures, with quantification, to support their claims. Similarly in Figure 4, the reported longitudinal chitin filaments are visible, but there are also some perpendicular arrangements, which undermine the argument that specific localization of kkv dictates ECM organization.*

We have analysed and quantified KkvGFP localization in L3 larvae, where it is easier to analyse and can provide higher magnifications. In addition, we have inserted another embryonic image in a new supplemental figure (Figure 3—figure supplement 1). However, for technical reasons we were not able to visualize kkvGFP and actin localization in the same images. Thus, as an alternative and due to the overlap between actin fibers and taenidia folds, we analized KkvGFP and chitin in the same images and found that more Kkv dots overlap with the taenidia and that the kkv dots associated with the taenidia are larger than the Kkv dots not overlapping with the taenidia. We have changed the text accordingly.

In Figure 4 the horizontal lines observed are an artefact from the SPE confocal laser and image acquisition on this channel. Sometimes we detect a fragmentation of the confocal image in an horizontal striped pattern when the image is acquired using a Leica SPE confocal due to the high magnification and low frequence required. In Figure 4 it might be more clear due precisely to its magnification and is absolutely unrelated to the chitin signal.

*4) The authors need to be exact in their discussion of the a-cat mutant phenotype. Since a-cat is not an integral junctional adhesion protein, the presented data do not show the cell-cell junctions are key players ECM patterning. The authors could bolster the evidence for involvement of adherens junctions by examining E-cad mutants, or demonstrating that adherens junction formation is disrupted in a-cat mutants. Alternatively, the existing evidence could be discussed in terms of an apparent requirement for linkage actin cytoskeleton to adherens junction complexes. The authors should also address concerns that while a-cat mutants clearly have disrupted ECM patterning near cell junctions, there still appears to be some registration of the taenidial ring between cells in the a-cat mutants. The authors should comment on this, and consider alternative models for the importance of kkv localization.*

We have analysed adherens junctions by means of DE-Cad and found that in all α-catenin downregulated tracheal tubes, we detected the phenotype and also a strong DE-Cad downregulation (n=8 embryos, no detectable DE-Cad staining). We have inserted a new panel in Figure 9—figure supplement 1 and made changes to the text to incorporate these data.

We have also added a new panel in Figure 9, showing a higher magnification of a junctional region, where it is clear there is an interruption of the taenidial continuity.

It is striking that despite the role of cell junctions in taenidial continuity, these are still placed perpendicular to the tube length. This suggests, as It was first suggested by Matusek et al., that tracheal cells are able to sense a global orientation cue and align their actin bundles appropriately, despite having their continuity disrupted. We have added reference to this in the manuscript.

*5) While models are necessarily imprecise abstractions of data, the reviewers feel the presented model in Figure 11 over interprets multiple aspects of the presented data, including the colocalization of kkv with chitin, the importance of src in the feedback loop to actin ring organization and the specificity of tal and Blimp-1 acting only on actin and chitin. The authors should also check the signs of the arrows in the model, as well as the localization of src (which is shown as an extracellular protein), as there appears to be errors in the original Figure 11. It is also unclear that the figure shows a time series.*

We have redrawn our model following the referees’ suggestions. Most importantly, we have now represented Src-phosphorylated, rather than just Src. We believe this new model is a more accurate description of our hypothesis.

[Editors' note: further revisions were requested prior to acceptance, as described below.]

*We agree that you have made many important changes, including to the model. Unfortunately, the central issue about the involvement of src has remained unanswered. For the manuscript to be considered for publication, please address the following two points. The first by performing one more set of experiments and the second by addressing the issue in the revised manuscript.* 1) Requires new experiment

*[…] For the paper to reach the standards of* eLife

*, as stated in the original review, it is critical to establish a role Src in actin ring formation. If the current model is correct that Src is not required for actin ring formation, but that Src activity must not be too high, then it should be the case that a Blimp-1;src42 or kkv;src42 double will have intact actin rings because there is no Src activity to disrupt them. The reviewers appreciate that it is quite possible, perhaps even likely, that ECM will have a more complex role in actin ring formation than keeping Src activity low, and thus that the proposed double mutants will still have abnormal actin rings, but it is essential for this paper to demonstrate an actual role for Src in actin ring formation. The proposed double mutant experiments are not presented as required experiments, but rather as possible experiments that could, using loss-of-function approaches, demonstrate a role for Src in actin ring formation. Alternative experiments would be acceptable, as long as they demonstrate a biologically meaningful role for Src in actin ring formation.*

We have made a *Src42A;Blimp-1* double mutant and analysed its taenidial/actin ring phenotypes. The double mutant displays a clear rescue of the taenidial alignment, while keeping other *Blimp-1* mutant features, confirming the biological role of Src42A overactivation in the disruption of taenidial/actin ring organization. We have included a new panel in Figure 10 and added changes to the manuscript text (subsection “The chitin apical extracellular matrix influences the levels of Src42A phosphorylation”, second paragraph).

*Parenthetically, it is possible that the anti-p-src stainings in the paper reflect src activity. It is also possible that they also reflect p-Abl activity (M. Tamada et al., Dev. Cell, 2012, 309-319). Please include controls to show that the staining is eliminated in src mutants.*

As stated in our previous letter the p-Src antibody reported in the Zallen paper (pY416) is not the same as the one we use (pY418) as we specify in the Materials and methods section. Indeed, the antibody we use was already assayed in homozygous embryos for a *Src42A* mutation that leads to an amino-acid exchange in the ATP-binding site (G257R) of the kinase domain and in these embryos the pSrc signals were abolished (Förster and Luschnig, 2012). In addition, we have done the stainings requested and added them to Figure 10—figure supplement 1.

2) Modifications to the text

*The last paragraph of the subsection “Tracheal actin rings relate to the spatial distribution of the chitin synthase” suggests kkv does not align with actin in tal/pri mutants. However, the actin patterns look strikingly similar to the taenidial patterns, and the last paragraph of the subsection “Chitin deposition and actin bundling contribute to proper taenidial fold organization “says there is "close interplay between actin and chitin in both tal/pri and Blimp-1 mutants". If kkv is not aligned with the actin in these mutants, this result actually undermines the argument that actin organizes kkv, which in turn produces an organized chitin pattern. Please clarify and explain in the text.*

We never meant to say we could not detect Kkv dots in tal/pri mutants. What we meant with this sentence was that in tal/pri mutants Kkv “lines” were not perpendicular to the tube length as in the wt. We are sorry about the lack of clarity and we have changed the manuscript (subsection “Tracheal actin rings relate to the spatial distribution of the chitin synthase”, last paragraph) in order to make it clearer.

We have also addressed the other comments by making amendments to the text and figure legends.